



# The Zurich Low-cost CO$_2$ sensor network (ZiCOS-L): data processing, performance assessment and analysis of spatial and temporal CO$_2$ dynamics

Luce Creman[1], Stuart K. Grange[1,a], Pascal Rubli[1], Andrea Fischer[1], Dominik Brunner[1], Christoph Hueglin[1], Lukas Emmenegger[1], and Leonie Bernet[1]

[1]Empa, Swiss Federal Laboratories for Materials Science and Technology, Laboratory for Air Pollution/Environmental Technology, Überlandstrasse 129, 8600 Dübendorf, Switzerland
[a]now at: Climate and Environmental Physics, Physics Institute, University of Bern, Sidlerstrasse 5, 3012 Bern, Switzerland

**Correspondence:** Lukas Emmenegger (lukas.emmenegger@empa.ch); Leonie Bernet (leonie.bernet@empa.ch)

**Abstract.**

As part of the ICOS Cities project, a network of low-cost NDIR (non-dispersive infrared) CO$_2$ sensors was set up across the city of Zurich (Switzerland), known as ZiCOS-L. The network consists of 56 sites with paired low-cost sensors spread out over the urban area of Zurich. This publication focuses on the period from August 2022 to July 2024. The sensors require in-field training for model calibration before deployment and further post-processing steps to account for drift and outliers. After data processing, the hourly mean root mean squared error (RMSE) was 13.6±1.4 ppm and the mean bias 0.75±1.67 ppm when validated against parallel reference measurements from the mid-cost sensor network ZiCOS-M. CO$_2$ concentrations were highly variable with site means in Zurich ranging from 438 to 465 ppm. These differences can largely be explained by the nearby surroundings, with vegetation, traffic density and human activity being dominant factors while altitude and distance from the city centre had a minor effect. Vegetation (mainly grassland) amplified the morning concentration in summer by up to 20 ppm due to ecosystem respiration, while heavy traffic increased the morning rush hour concentration by 15 ppm. Human activity was shown to locally enhance CO$_2$ concentrations during two public events. Despite its lower measurement accuracy, the ZiCOS-L network enables the study of concentration dynamics at a spatial and temporal scale that could only be achieved at much higher cost with mid-cost or high-precision instrumentation. The observations generated by ZiCOS-L will be further used in ICOS Cities activities to validate CO$_2$ emission inventories with inversion modelling systems.



## 1 Introduction

The Earth's atmosphere has been exposed to approximately 1 degree of warming compared to pre-industrial times
due to anthropogenic emissions of greenhouse gases (GHGs) such as carbon dioxide ($CO_2$) (IPCC, 2023b). Cities
are by far the largest contributors to climate change (Duren and Miller, 2012). While urban areas only make up
2 % of the earth's surface, they are responsible for up to 70 % of global anthropogenic emissions (IPCC, 2023a;
United Nations, 2011). It is for this reason that many cities are taking action by joining the "race to zero" campaign,
with the goal of reducing their GHG emissions to net zero by 2050 or earlier (C40 Cities, 2025). To achieve such
targets, it is important to check the efficiency of policy plans, which can be done by use of a monitoring network.

Monitoring $CO_2$ concentrations in urban areas is particularly challenging due to large spatial and temporal
variability of emissions and the complex topology of the urban landscape. Cities usually rely on annual bottom-
up emission inventories based on a combination of activity data, emission factors, and behavioural assumptions.
These inventories can be verified using top-down inversion modelling, which need ground-level observations as an
input.

Until recently, observations were usually made by a small number of expensive high precision instruments, usually
on tall towers measuring background concentrations. Despite the high accuracy of such measurements, they only
give limited insights into spatial variability of concentrations at a local scale (World Meteorological Organization,
2024; Shusterman et al., 2016; Hernández-Paniagua et al., 2015). This highlights the need for lower-cost real-time
$CO_2$ monitoring systems that measure at the scale of local emissions. As a response, the availability of low-cost
atmospheric trace gas monitoring tools is on the rise (World Meteorological Organization, 2024). Of course, lower
cost limits data quality, so a trade-off should be made between cost and quality, depending on the purpose of use
(World Meteorological Organization, 2024). Low-cost observations need extensive calibration and data processing,
and uncertainties and limitations of the measurements must be considered during the interpretation of the data.
However, the advantage of low-cost sensors is that they can be deployed at higher density, enabling $CO_2$ concentration
patterns to be resolved at a much smaller scale (Castell et al., 2017).

The definition of a low-cost sensor differs between users. In Zurich, lower-cost sensors are therefore divided into
two tiers: low- and mid-cost. The Zurich ICOS Cities Mid-cost (ZiCOS-M) network contains sensors which are at
an order of 7–14 times lower price than a high-cost, high-precision cavity ring-down analyser (Grange et al., 2025;
Shusterman et al., 2016). Another example of a network of similar cost is the Berkeley Environmental Air-quality
& $CO_2$ Observation network (BEACO$_2$N), operating in the San Francisco Bay area since 2013 (Shusterman et al.,
2016, 2018; Delaria et al., 2021). BEACO$_2$N has recently extended towards Los Angeles and Providence (United
States) and Glasgow (United Kingdom) (Kim et al., 2025). Additionally, as part of the ICOS Cities project, a
mid-cost network has been deployed in Paris (Lian et al., 2024) and Munich (Aigner et al., 2024).

In this study, we present a network at another order of magnitude lower cost which we therefore refer to as the
Zurich ICOS Cities Low-cost network (ZiCOS-L). Sensors of the same price point have previously been installed



in the Carbosense network which extended across the whole of Switzerland (Müller et al., 2020). ZiCOS-L is an extension to the Carbosense network, with a local focus on the city of Zurich comprising 56 valid $CO_2$ monitoring sites distributed over an urban area of around 170 km$^2$. A $CO_2$ monitoring network using this price-class of sensors

in such a high density is so far unprecedented, making ZiCOS-L a valuable addition to the existing range of low- and mid-cost $CO_2$ monitoring networks by providing insights into $CO_2$ dynamics at a high spatial and temporal resolution.

## 1.1 ICOS Cities

This work is part of the ICOS Cities project, funded by the European Union's Horizon programme (https://www.
icos-cp.eu/projects/icos-cities, last access: 16.06.2025). The project supports the European Green Deal by developing and testing tools and services for cities, supporting their climate action plans to reduce GHG emissions. The project entails 15 cities in Europe of which three cities of different sizes have been selected as pilot cities. From large to small, the pilot cities are Paris, Munich and Zurich, and they are used to test the effectiveness and feasibility of various monitoring techniques. A network similar to ZiCOS-L has been installed in the medium-sized city of Munich
(Kühbacher and Chen, 2024). Differences and similarities between these two networks can give valuable insights into the usefulness and usability of low-cost $CO_2$ networks to other cities.

## 1.2 Objectives

The aim of this work is to evaluate the potential of a dense low-cost $CO_2$ monitoring network in a European city of relatively small size. The objectives are to:

1. describe the network design, sensor deployment and data processing,

2. evaluate measurement performance of the network,

3. present spatial variability and temporal dynamics that can be detected by the network.

The first objective can aid other cities with the intention of setting up a similar monitoring network because most of the data processing steps can be applied to other network settings. The second objective is important for the
interpretation of the data, with a specific focus on inversion modellers who can potentially use the observations in their models to validate emission inventories. The last objective can be of interest to city stakeholders to identify $CO_2$ hotspots and sources and adapt their policies to reduce GHG emissions accordingly.

The goal of the network is not to produce high quality data to accurately resolve regional patterns, but rather to give insights into short-term and highly local $CO_2$ dynamics close to emission sources and sinks.

This paper is a follow-up on the ZiCOS-M publication on Zurich's mid-cost $CO_2$ sensor network presented in Grange et al. (2025). As highlighted earlier, the low-cost network largely depends on the mid-cost network, and the two papers therefore go hand-in-hand, following a similar structure.





## 2 Methodology

### 2.1 Network design

Zurich's ICOS Cities $CO_2$ sensor network (ZiCOS) is composed of a total of 234 sensors of different price points that cover the urban area of the city of Zurich (Fig. 1). The network is an extension of the national CarboSense project described by Müller et al. (2020) with a local focus on the urban area of the city of Zurich.

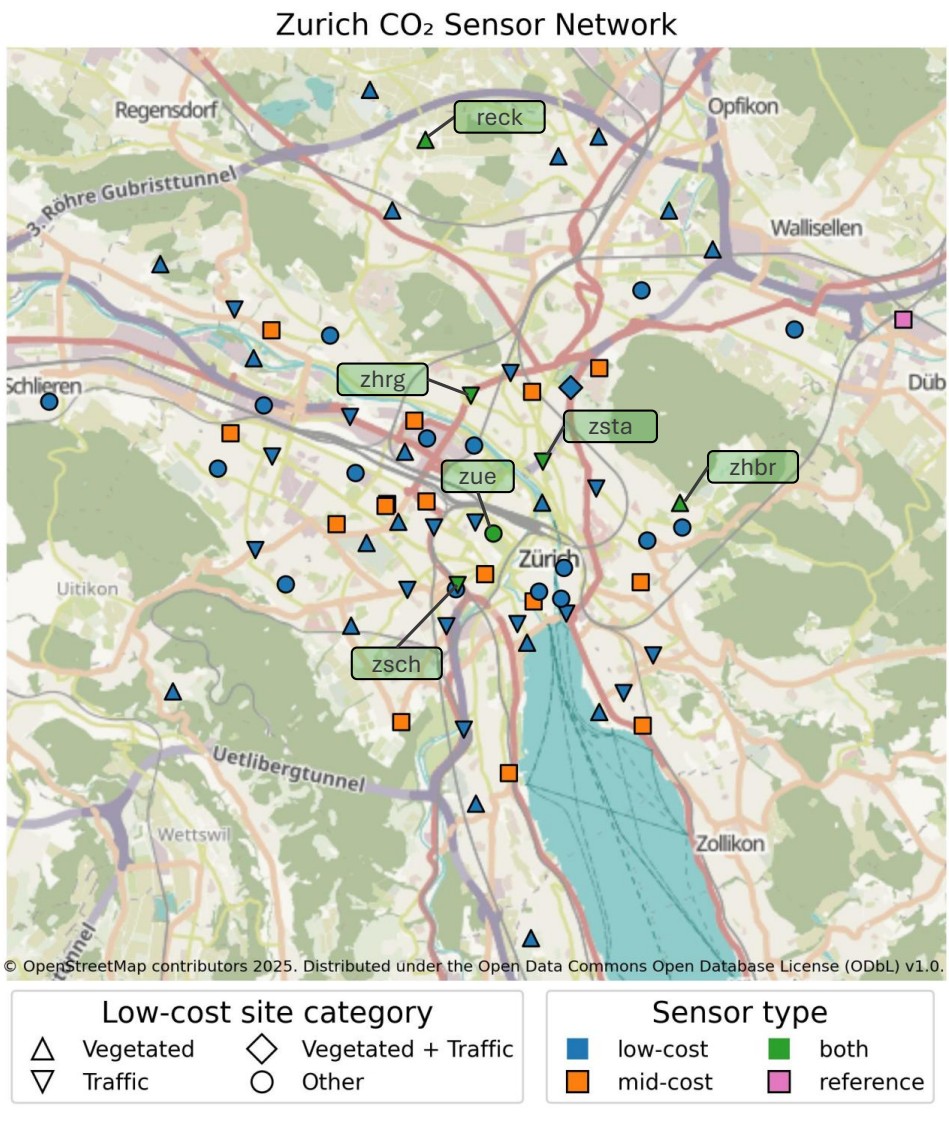

**Figure 1.** A map of the ZiCOS network. The colors indicate the sensor type. At locations with low-cost sensors (both low-cost (blue) and co-located (green) sites), the marker indicates the site category.



Paired low-cost sensors are deployed at 61 different sites, of which 56 sites produced valid observations in our study period. The low-cost sensors are generally mounted to light poles or traffic signs and thus located in street-level environments. Together, these sensors make up Zurich's ICOS Cities low-cost $CO_2$ sensor network (ZiCOS-L). Additionally, there is a mid-cost sensor network (ZiCOS-M) deployed, consisting of 21 sensors in the urban area, which are installed in climate-controlled indoor environments and generally sample at rooftop level. A full description of the ZiCOS-M network can be found in Grange et al. (2025). There are six sites where the two networks overlap, with parallel measurements of a mid-cost sensor and a low-cost sensor pair. At these locations, both mid-cost and low-cost sensors are installed at near-ground level. Lastly, Empa-Dübendorf's air quality monitoring site acts as the reference site hosting a high-cost and high-accuracy cavity ring-down spectroscopy (CRDS) gas analyser (Picarro G2302).

Deployment of the ZiCOS-L network started in June 2022, and the full network was running from mid-July 2022 onward. Currently, the network is still in operation and is expected to continue for a few more years. This study focuses on the two-year period between beginning of August 2022 and end of July 2024.

All sites were carefully selected to capture as much of the spatial variability of the city as possible. Site details are documented, including exact coordinates, latitude, aerial photographs and a description of the surroundings (Appendix A). Consequently, the sites are classified based on potential sources and sinks in their nearby surroundings that can affect the concentration. This classification will be used to analyse the effect of vegetation, traffic and human activity on the local concentration.

For traffic, the sites are divided into four groups of traffic intensity: 'heavy', 'moderate', 'light' and 'none'. A site will be referred to as a traffic site when the traffic intensity is either moderate or heavy. For vegetation, a distinction is made for densely or sparsely vegetated sites. In turn, the densely vegetated sites are split up into three vegetation types: 'grassland', 'forest' and 'mixed', whereas the sparsely vegetated sites are split up into 'some', 'single tree' and 'none'. The group 'single tree' refers to a site where there is little vegetation, but the sensor is mounted right next to a single tree which can affect the measured concentration. A site is defined as vegetated when the vegetation is dense (so, belonging to 'grassland', 'forest' or 'mixed'). Thanks to this classification framework, a site can be both vegetated and a traffic site at the same time. An example of the classification of four monitoring sites is depicted in Table 1. Lastly, to examine the effect of human activity, two individual sites were selected: Langstrasse, a busy nightlife street, and Limmatquai, a site in the city centre with many daytime visitors.

## 2.2 LP8 $CO_2$ sensor

The low-cost sensor units used in this network are manufactured by Decentlab GmbH and consist of a SenseAir LP8 $CO_2$ sensor and an ancillary Sensiron SHT21 Temperature ($T$) and Relative Humidity ($RH$) sensor ($\pm\,0.3\,°C\ T$, $\pm\,2\,\%\ RH$). The sensor unit is identical to the one presented and described by Müller et al. (2020). Both the LP8 and the SHT21 sensors are mounted close to the opening to ensure quick response times (Fig. 2b).




**Table 1.** Photographs and metadata of four low-cost $CO_2$ monitoring sites in Zurich. Sensor positions are circled in red.

|  | **Stampfenbachstrasse** | **Katzensee** | **Irchelpark** | **Schule Seefeld** |
|---|---|---|---|---|
|  |  |  |  |  |
| **Area** | Urban | Rural | Suburban | Suburban |
| **Vegetation** | Some | Grassland | Forest | Single tree |
| **Traffic** | Heavy | None | Moderate | Moderate |

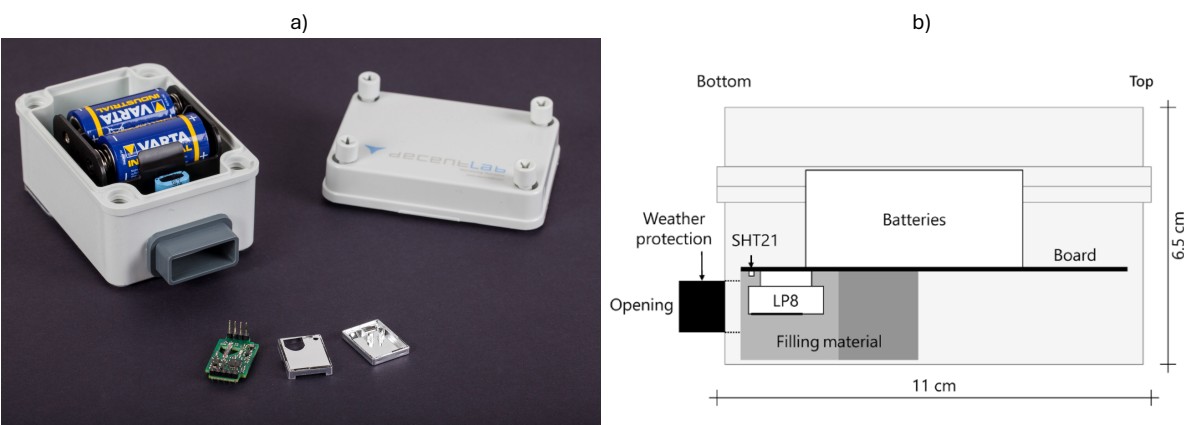

**Figure 2.** $CO_2$ (a) The integrated low-cost sensor unit (back) and the SenseAir LP8 $CO_2$ sensor and LoRaWAN data transmission technology (front) and (b), a schematic overview of the interior of the sensor unit. The LP8 and SHT21 sensors are mounted close to the opening. Picture: Müller et al. (2020)



The LP8 sensor is not actively ventilated and relies on diffusive processes to detect changes in ambient $CO_2$ concentrations. The accuracy of the $CO_2$ measurement based on factory calibration is $\pm 50$ ppm with an additional $\pm 3\%$ of reading, which is insufficient to capture the typical variability of $CO_2$ in ambient air. The sensor unit runs on the power of two 'C' alkaline batteries, which ensure sensor operation of around five years without replacement

(Müller et al., 2020). Decentlab recently brought out a newer version of this unit under the name DL-LP8P which additionally includes an atmospheric pressure ($P$) sensor and is also commercially available (Decentlab, 2025).

The per-unit price of the SenseAir LP8 low-cost $CO_2$ sensor is around EUR 50. However, with additional costs for deployment, low-power integration, maintenance and processing, the cost is estimated at EUR 500 per sensor. In comparison to the mid-cost sensors presented in Grange et al. (2025) costing around EUR 5 000–10 000, the low-cost

sensors are at a significantly lower price point.

### 2.3 Data transmission, processing and storage

Data is transmitted using a Low Power Wide Area Network (LPWAN) named LoRaWAN (LoRaWAN® Aliance, 2015). The data transmission (data packet en- and decoding, database management) is provided by Decentlab as a service. Measurements are made approximately every minute and averaged into 10-minute intervals onboard the

LP8 sensor unit, after which they are transmitted. Due to the low-power integration, no data storage is possible on-board, therefore, if data transmission fails, the data is lost. However, the long lifetime of the sensors at their monitoring sites can largely be attributed to the low power transmission made possible by LoRaWAN, making it a key element for the ZiCOS-L network.

Measurement data is hosted on Decentlab's InfluxDB database and can be accessed through an API. Scheduled

tasks, running on Empa local machines, maintain regular (*i.e.* daily) data processing as will be described in the following section. For local storage during computations, an Empa-internal MySQL database serves as temporary storage and to store metadata such as information concerning location, deployment and sensor IDs. Processed data is then re-uploaded to Decentlab's InfluxDB for long-term storage and allowing simple data visualisation using Grafana Dashboards (Grafana Labs, 2025). The same architecture for data transmission, processing and storage was used for

the ZiCOS-M network as described in Grange et al. (2025).

### 2.4 Data processing

In preparation for field deployment, all sensors were first tested in a climate chamber (Appendix C) and then co-located in ambient field conditions at the Dübendorf-Empa reference air quality monitoring site for varying timespans within a one-month period. Subsequently, the low-cost sensors were deployed in the field without reference gases and

were generally left without maintenance until a failure or a clear issue was observed, thus adhering to the low-cost requirements.

The sensors are known to be dependent on environmental factors and susceptible to temporal drift (Cai et al., 2025; Delaria et al., 2021; Müller et al., 2020). These drawbacks were managed after deployment with post-processing





procedures to improve the sensor's accuracy. The methods we propose are thus entirely based on *in situ* corrections after the co-location period at the reference site. The post-processing was done in a sequential stepwise manner, with each step increasing the accuracy of the observations. Firstly, the reported measurement is converted to dry air mole fraction. Secondly, the measurements are calibrated using a multi-linear regression model, after which they are corrected for drift. Finally, the data is filtered for conditions during which the measurements might be compromised by the environment. The next sections will elaborate further upon each of the post-processing steps that were carried out.

### 2.4.1 Water dilution correction

The LP8 sensors report $CO_2$ in moist air. However, we prefer to express $CO_2$ as dry air mole fraction as this quantity is preserved during atmospheric transport and phase changes. The reported $CO_2$ was converted to dry air mole fraction using the amount of water vapour in the air based on $RH$ and the $T$ dependent saturation vapour pressure following the August-Roche-Magnus formula, an approximation of the Clausius-Clapeyron equation (Alduchov and Eskridge, 1996). $T$ and $RH$ data are measured by the Sensirion SHT21 sensor in the sensor unit of each low-cost sensor, so they are local observations. This conversion will be referred to as the water dilution correction. Throughout this article, we will refer to the dry air mole fraction as 'concentration'.

### 2.4.2 Calibration models

The factory calibration provided by the manufacturer is insufficient for the accurate monitoring of ambient air (accuracy of $\pm 50\,\text{ppm}$) and thus requires additional calibration. For this step, a collection of nine calibration models, including five multivariate linear regression models, two Lambert-Beer models and the two more complex models (full and restricted) used in the previous Carbosense project as described by Müller et al. (2020), were tested. Each model was once trained on the climate chamber data and tested on the reference gas analyser observations at Dübendorf, and once trained on the reference gas analyser and tested on both the gas analyser at Dübendorf before deployment and the mid-cost sensors in the field after deployment.

As described above (Sect. 2.2), each sensor unit also measures $T$ and $RH$, making them the most obvious features for the regression models. Additionally, ambient pressure ($P$) measured by the reference gas analyser or mid-cost sensors was used. However, since $RH$ itself is $T$ dependent, we decided to additionally test a model using specific humidity ($q$) (derived from $RH$ and $P$).

The tested linear regression models started from a basic model using $CO_2$ as the only feature, which was stepwise extended by introducing new features in the following order: $T$, $RH$ or $q$ and finally $P$. The Lambert-Beer and the models proposed by Müller et al. (2020) used the sensors' raw infrared measurement, while the (multi-) linear models used $CO_2$ mole fraction after the water dilution correction.

Comparison of the different model training and testing methods demonstrated that training with climate chamber data was not representative of field conditions, which is in line with findings of many other studies (Cui et al., 2021;





Malings et al., 2019; Topalović et al., 2019; Castell et al., 2017). Consequently, the calibration models were finally trained on 3–12 weeks of ambient reference gas analyser observations at Dübendorf-Empa before deployment and tested on mid-cost sensor observations at their field location right after deployment. Note that the models could 190 only be tested for the paired low-cost sensors that are co-located in the field with a mid-cost sensor, and that these mid-cost sensors are an imperfect reference with a known uncertainty of about ±1 ppm (Grange et al., 2025).

The model formulation that consistently performed best was the multi-linear model that included $CO_2$, sensor $T$, and sensor $RH$. It should be mentioned that the model using $q$ instead of $RH$ performed equally well, but since $RH$ is measured directly by the sensor, this model was preferred. The calibrated concentrations ($[CO_2]^{cal}$) are calculated 195 from the reported concentration ($[CO_2]^{rep}$) following Equation 1:

$$[CO_2]^{cal} = [CO_2]^{rep} + \alpha_{[CO_2]^{rep}} \cdot [CO_2]^{rep} + \alpha_T \cdot T + \alpha_{RH} \cdot RH \tag{1}$$

in which $\alpha_{[CO_2]^{rep}}$, $\alpha_T$ and $\alpha_{RH}$ are the sensor-specific correction coefficients for $[CO_2]^{rep}$, $T$ and RH respectively. Each sensor thus has a coupled calibration model with sensor-specific coefficients.

### 2.4.3 Outlier removal

After calibration, the calibrated low-cost sensors' observations $[CO_2]^{cal}$ were passed through a Hampel identifier algorithm to identify measurement spikes. The Hampel identifier is a windowed median and median absolute deviation (MAD) scale identifier (Pearson et al., 2016). The identifier evaluates a value's distance from the median, and if it is larger than the MAD scale implied, it will be replaced by the median value of the window. The Hampel identifier is a 'despiking' algorithm that effectively removes outliers and reduces noise. For the low-cost sensors, a 205 large window of 20 hours was applied to invalidate large spikes which represented sensor measurement errors. The despiking was done at 10-minute level, and after aggregation back to hourly level 16 % of data was invalidated. A similar but more strictly tuned Hampel identifier was also used for the mid-cost sensors' observations to identify possible local contamination events (Grange et al., 2025).

### 2.4.4 Drift correction

The low-cost sensors often suffered from severe drift over time. The drift primarily manifested in changes in the sensors' baseline concentrations and was, for some sensors, up to one ppm per day. This caused an issue where the sensor-calibration model system (Sect. 2.4.2) was not stable over time, requiring an additional adjustment accounting for the network baseline. A large number of network calibration methods have been documented and used for gas sensing sensor networks (Weissert et al., 2020, 2019; Delaria et al., 2021; Müller et al., 2020; Bigi et al., 2018; Miskell 215 et al., 2018, 2016; Shusterman et al., 2016). Most of these methodologies rely on a single or a few reference sites with high accuracy observations that are robust against drift. For example, Müller et al. (2020) used wind as a proxy for homogenous conditions and then rely on three remote Picarro sites, depending on the wind direction, to





correct the drift during these conditions. In our case, we took advantage of the high spatial resolution of the mid-cost network for more local references. Therefore, we present a novel approach for drift correction using local mid-cost

measurements.

To correct for drift, we subtract a time series of offsets from the low-cost observations ($[CO_2]^{LC}$):

$$[CO_2]_{corrected}^{LC,drift}(t) = [CO_2]^{LC}(t) - \Delta[CO_2](t) \qquad (2)$$

The offset $\Delta[CO_2](t)$ is the difference between the low-cost observation and the reference network baseline.

The network baseline refers to a time series of reference minimum concentrations, in our case from the coupled

mid-cost sensor network. The mid-cost sensors are equipped with reference gas cylinders which are used to internally calibrate the measurements, resulting in more accurate observations compared to the low-cost sensors. The uncertainty of the mid-cost sensors is an RMSE of 1 ppm with a bias of -0.09 ppm (Grange et al., 2025), and can thus be considered as an in-field reference. Full discussion of the mid-cost sensor performance can be found in Grange et al. (2025).

Since not all low-cost sensors are co-located with a mid-cost sensor, the network baseline first needs to be derived from a spatial interpolation of mid-cost baselines, so an estimate can be extracted at each low-cost sensor site. This was done as follows: The daily minimum concentration was calculated for 22 sites across Zurich city (21 mid-cost sensor sites and the reference gas analyser site). Daily minima are defined as the minimum hourly concentration ($[CO_2]_{min}$) experienced between 14:00 and 17:00 (local time). These daily minima were interpolated using an inverse

distance weighting algorithm to build daily minimum concentration surfaces with a resolution of 500 m for the spatial extent in which the low-cost sensors are deployed, which corresponds to an area of about 170 km$^2$. As a heuristic, the second lowest $CO_2$ concentration of the 22 mid-cost sites was used as the surface's baseline to omit potential anomalies. An example of such a daily minimum concentration surface is shown in Fig. 3. This surface represents the network baseline $CO_2$ concentration at a given day.

A drawback of this method is that the (co-located) mid-cost sensors are also used for validation, making the processed data dependent on the validation data. Consequently, the surface interpolation was also computed by withholding the six co-located sites as a check. The surfaces with withheld sites returned virtually identical values. Therefore, we concluded that the initial interpolated surfaces are safe to use for further data processing and the surface with withheld sites was not used.

Initially, $\Delta[CO_2]$ is made up of two components: a variable daily offset ($\Delta[CO_2]^{daily}$) and a static starting increment ($\Delta[CO_2]^{start}$):

$$\Delta[CO_2]^{init}(t) = \Delta[CO_2]^{daily}(t) - \Delta[CO_2]^{start} \qquad (3)$$

where $t$ is time in days.





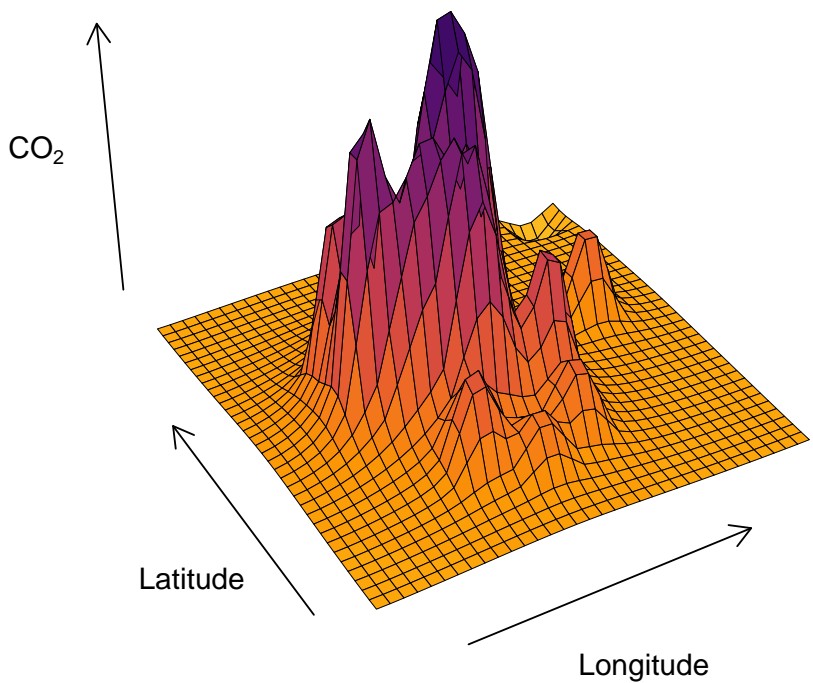

**Figure 3.** Daily $CO_2$ minimum naive concentration surface for December 20, 2022. The domain of the surface is the low-cost sensor network extent $(170\,\mathrm{km}^2)$ and concentrations for this day were especially variable among the sites.

Firstly, a time series of daily offsets $(\Delta[CO_2]^{\mathrm{daily}}(t))$ is computed by taking the difference between the daily

minimum low-cost observation $([CO_2]_{\min}^{\mathrm{LC}})$ and the baseline estimate from the mid-cost network $([\widehat{CO_2}]_{\min}^{\mathrm{MC}})$:

$$\Delta[CO_2]^{\mathrm{daily}}(t) = [CO_2]_{\min}^{\mathrm{LC}}(t) - [\widehat{CO_2}]_{\min}^{\mathrm{MC}}(t) \tag{4}$$

The baseline estimate $[\widehat{CO_2}]_{\min}^{\mathrm{MC}}$ is computed at each low-cost site by extracting the daily minimum mid-cost concentration from the interpolated surface at the spatial location of the low-cost site for each day of the study period.

In addition to the daily offsets, a key difference in the placement of the low-cost and mid-cost sensors needs to

be taken into account: mid-cost sensors are generally installed on rooftops, while low-cost sensors are installed at street-level. Therefore, a baseline estimate based on mid-cost sensors is limited in its representativeness for the low-cost network. Even at the same site (same spatial location but different height), a low-cost sensor can be expected to experience different (usually higher due to closer proximity to emission sources) concentrations than the mid-cost sensor. To solve this issue, we introduce $\Delta[CO_2]^{\mathrm{start}}$ which represents this difference at the start of deployment.

Since the low-cost sensors were first co-located with a reference gas analyser for calibration at the testing site (Dübendorf-Empa) and deployed at their field sites immediately after, it was assumed that the sensors' calibration was "correct" at the start of the deployment. Accordingly, the starting increment was calculated by taking the average value of the difference between the low-cost observation and the baseline estimate during the second and



third day of field deployment:

$$\Delta[CO_2]^{start} = \text{mean}_{t \in \{2,3\}} \left( [CO_2]_{min}^{LC}(t) - [\widehat{CO_2}]_{min}^{MC}(t) \right) \tag{5}$$

The starting increment and daily offsets together made up the initial offset coefficient (Equation 3), which was slightly smoothed using a three-day rolling mean, and finally subtracted from the calibrated low-cost concentration (Equation 2). However, after several months of monitoring, it was determined that the starting increment logic was too simple to capture the low-cost sensor sites' $CO_2$ dynamics. The initial offset was inadequate to represent the sensors' behaviour during the entire monitoring period, including different seasons. This led to persistent biases for some sensors. Therefore, another layer on top of the correction steps above was needed, to create an additional component of the offset coefficient ($\Delta[CO_2]^{homog}$) during homogeneous network conditions:

$$\Delta[CO_2](t) = \Delta[CO_2]^{daily}(t) - \Delta[CO_2]^{start} + \Delta[CO_2]^{homog}(t) \tag{6}$$

Homogeneous conditions were defined as days with small horizontal gradients in $CO_2$ concentration. Such conditions generally occur when atmospheric dispersion and transport are high, *i.e.*, during windy conditions. In contrast to the Carbosense network, in which homogenous conditions were identified based solely on wind speed (Müller et al., 2020), we directly assessed the homogeneity by considering the network gradient. The network gradient ($G(t)$) was assessed between daily minima at Lägern Hochwacht ('laeg'), a gas analyser background site located just outside the urban area (Appendix B), and Schimmelstrasse ('zsch'), a co-located mid-cost and low-cost sensor site in the city next to a busy road and therefore with generally high concentrations (Fig. 1):

$$G(t) = [CO_2]_{min}^{MC,zsch}(t) - [CO_2]_{min}^{GA,laeg}(t) \tag{7}$$

These two sites were selected because they consistently measured the lowest and highest concentrations in the network, and had continuous data availability throughout the study period.

A sensitivity analysis showed that network gradients were always higher in autumn and winter compared to summer and spring due to stronger emissions and lower dispersion. A fixed gradient threshold would thus lead to updates being spread irregularly throughout the year. To make updates more evenly spread, season-based afternoon $CO_2$ gradient thresholds ($\theta_s$) were applied:

$$\theta_s = \begin{cases} 15\,\text{ppm}, & \text{winter} \\ 9\,\text{ppm}, & \text{spring} \\ 9\,\text{ppm}, & \text{summer} \\ 14\,\text{ppm}, & \text{autumn} \end{cases} \tag{8}$$

To ensure that the updates were only executed when the low-cost sensors were unlikely to be compromised by environmental conditions, some thresholds were set. Cold ($T \le 2\,°C$), hot ($T \ge 28\,°C$), and/or humid days ($RH \ge$





95 %; defined between 08:00 and 19:00 local time) were classed as inappropriate as updating days. A homogenous day ($\mathcal{H}$) suitable for updates is thus defined as:

$$t \in \mathcal{H} \quad \text{if} \quad \begin{cases} G(t) < \theta_s \\ 2°C < T(t) < 28°C \\ RH(t) < 95\% \end{cases} \tag{9}$$

Once a reliable and homogeneous day was detected, the difference between the gas analyser at Dübendorf and
the low-cost observations after initial offset correction ($[CO_2]_{min}^{LC,init}$) was used to create $\Delta[CO_2]^{homog}$ at each homogeneous day. Similar to the previous components of the offset coefficient, this was done for the minimum afternoon (between 14:00 and 17:00) $CO_2$ concentration, but this time at 10-minute level:

$$\Delta[CO_2]^{homog}(\mathcal{H}) = [CO_2]_{min}^{LC,init}(\mathcal{H}) - [CO_2]_{min}^{GA,due}(\mathcal{H})$$

$$= ([CO_2]_{min}^{LC}(\mathcal{H}) - (\Delta[CO_2]^{daily}(\mathcal{H}) - \Delta[CO_2]^{start})) - [CO_2]_{min}^{GA,due}(\mathcal{H}) \tag{10}$$

Homogeneous days suitable for updates occurred approximately once per month and the resulting offset components
$\Delta[CO_2]^{homog}(\mathcal{H})$ were interpolated in time to get daily offset components ($\Delta[CO_2]^{homog}(t)$). Several sensitivity tests
with different time periods and anchor points showed that this approach led to the best predictions. Ultimately, the
low-cost baseline is adjusted using the final coefficient $\Delta[CO_2](t)$ (Equation 6), and the final drift-corrected low-cost
concentration is calculated according to Equation 2:

$$[CO_2]_{corrected}^{LC,drift}(t) = [CO_2]^{LC}(t) - \Delta[CO_2](t) = [CO_2]^{LC}(t) - (\Delta[CO_2]^{daily}(t) - \Delta[CO_2]^{start} + \Delta[CO_2]^{homog}(t)) \tag{11}$$

A schematic visualisation of the drift correction using the network baseline and the different offset components is
outlined in Figure D1.

### 2.4.5 Environmental filtering

According to the manufacturer, the LP8 sensor's operation is limited by high $RH$ and low $T$ (Sect. 2.2). It is thus
necessary to disregard systematic errors caused by sensor malfunctioning during low $T$ and high $RH$. The $T$ and $RH$
measured by the low-cost sensor unit (Sect. 2.2) can be used to filter out conditions that cause malfunctioning. Note
that sensor $T$ and $RH$ are not the same as ambient conditions. For example, moisture accumulation in the sensor
unit or exposure to direct radiation during the day can result in higher sensor $T$ or $RH$ than ambient conditions.
However, these are the conditions inside the sensor unit, to which the LP8 sensor is exposed.

An analysis of the relation between observed $CO_2$ and $T$ showed abnormal sensor behaviour below $0\,°C$ for all
sensors. Therefore, $0\,°C$ was chosen as the $T$ threshold below which observations are disregarded. For $RH$, however,
the relation differs significantly between individual sensors. This is in line with Müller et al. (2020), who found $RH$
thresholds ranging between 82–94 %. We disregarded observations as soon as $RH < RH_{trsh}$. In order to remove as
few observations as possible, it is essential to identify $RH_{trsh}$ as precisely as possible for each sensor individually.





Müller et al. (2020) determined these thresholds by comparing to a reference, and looking at the standard deviation of these residuals. However, using the standard deviation will also filter out some of the natural variability that is not driven by humidity effects. Another method is presented by Delaria et al. (2021), who used the slope of the difference between the 10th percentile of their Vaisala instrument observations and Picarro supersite to correct for
the Vaisala's $T$ dependency.

Alternatively, we propose a new method, in which the sensor-specific $RH$ threshold is determined based solely on the low-cost $CO_2$ observations, by assessing the (5-bin rolling) slope of its relation with $RH$. As Fig. 4a–c depict, this slope generally ranges between -2 and 2 ppm/% for most sensors. However, for most of the LP8 sensors (*e.g.* sensors 1156 and 1046 in Fig. 4), the measured $CO_2$ starts to rapidly increase with increasing $RH$ after a certain
$RH$ threshold, resulting in a slope well above 2 ppm/% (Fig. 4a and b).

A sensitivity analysis was performed by varying the slope threshold within a range of 1–8 with 0.5 ppm/% intervals. The point where the slope exceeds the slope threshold determines the $RH$ threshold. Observations during which the $RH$ was higher than the threshold were disregarded and resulting changes in error metrics were calculated. Finally, $RH_{trsh} = 4$ ppm/% was chosen as the optimal slope threshold resulting in good data quality with a rejection rate of
15 %. A distribution of the $RH$ thresholds of all low-cost sensors corresponding to this slope threshold is depicted in Fig. 4d.

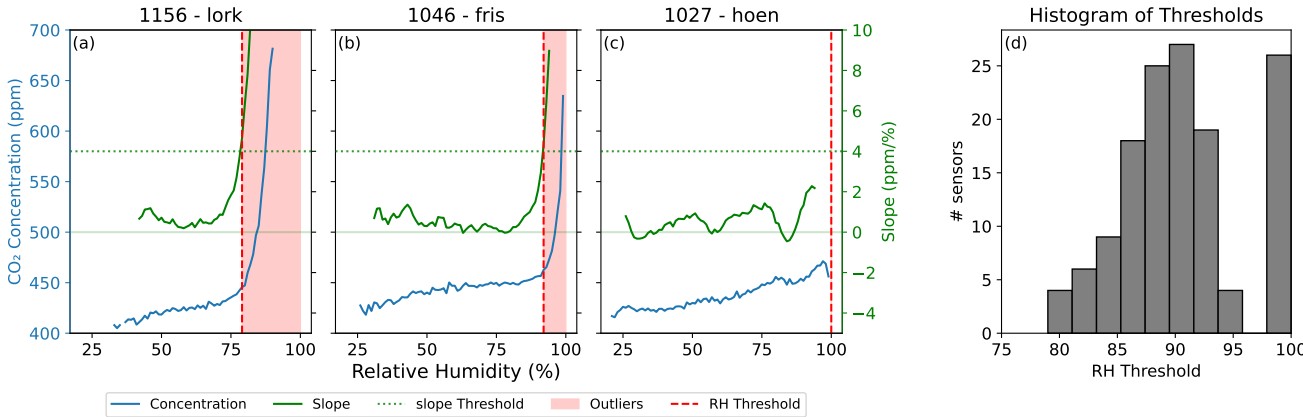

**Figure 4.** $CO_2$ observations of sensor units (a) 1156 at Loorenkopf, (b) 1046 at Friedhof Sihlfeld, and (c) 1027 at Meierhofplatz Höngg, binned in 1 % $RH$ intervals. The blue lines indicate the observed $CO_2$ concentration, and the green line shows the 5-bin rolling slope of the relation between the concentration and the $RH$. The dotted green line indicates the slope threshold, and the dashed red line the $RH$ threshold of the sensor. The red area shades the observations that are disregarded as a result of this threshold. Panel d) shows a histogram of the distribution of $RH$ thresholds of all low-cost sensors. When no threshold is needed, it is set to 100 % RH.





## 3 Results and discussion

After tandem deployment, the low-cost sensors were left in the field without any maintenance, only being replaced when they were malfunctioning. After around two years of measurements, the data was processed as described in

the previous section. In total 56 sites with reliable data between 1 August 2022 and 31 July 2024 were selected for further analysis. In this section we will first evaluate the measurement performance, and subsequently present spatial variability and temporal dynamics that were detected by the network.

### 3.1 Sensor performance

Since low-cost sensors are always deployed in tandem, the agreement between the two sensors already provides some

information on the accuracy of the sensors, and thus the data quality of the monitoring site. Figure 5 shows an example of how each data processing step described in the section above improves the agreement between the time series of two paired low-cost sensors at the same site. Notably, the low-cost sensors' reported and unprocessed $CO_2$ concentrations were shown to be unsuitable for use in the field (Fig. 5a). After ambient calibration, the low-cost sensors still demonstrated significant baseline drift (Fig. 5b) which required drift-correction at the field monitoring

sites (Fig. 5c). Finally, filtering for environmental thresholds removes some outliers and the final data of the two sensors are in satisfactory agreement.

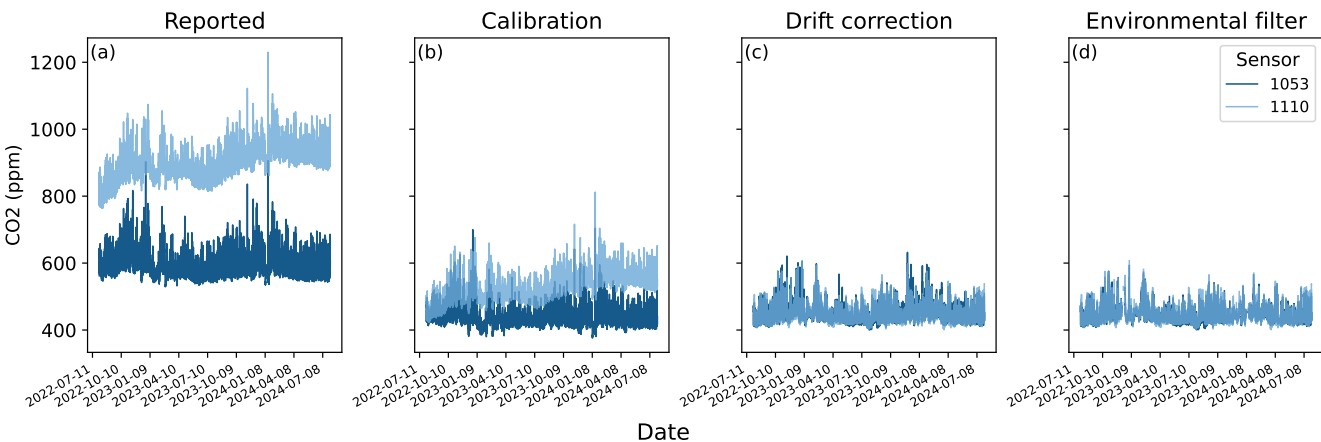

**Figure 5.** The time series $CO_2$ concentration after each processing step for the two paired low-cost sensors at the Platzspitz (PLAS) monitoring site. The improvement in agreement between the two sensors can be seen when moving from left to right, from the reported data to the final data.

As a more robust measure of accuracy, the final, calibrated, drift-corrected and cleaned $CO_2$ concentration was tested independently at six sites (with 17 low-cost sensors) in the network where both mid- and low-cost sensors were deployed in tandem (Fig. 1, green sites). The six sites consist of three traffic sites (zhrg, zsch and zsta), one urban



background site (zue) and two vegetated sites, one with grassland (reck), and one with forest (zhbr) (Appendix A).
To asses the performance of the low-cost sensors at these sites, the cylinder-adjusted mid-cost $CO_2$ observations
were used as a reference (Grange et al., 2025). Three statistical metrics were analysed: bias, root mean squared error
(RMSE) and the Pearson correlation coefficient ($r$-value). Figure 6 shows the improvement in data quality compared
to the mid-cost sensors after each processing step at all co-located sites.

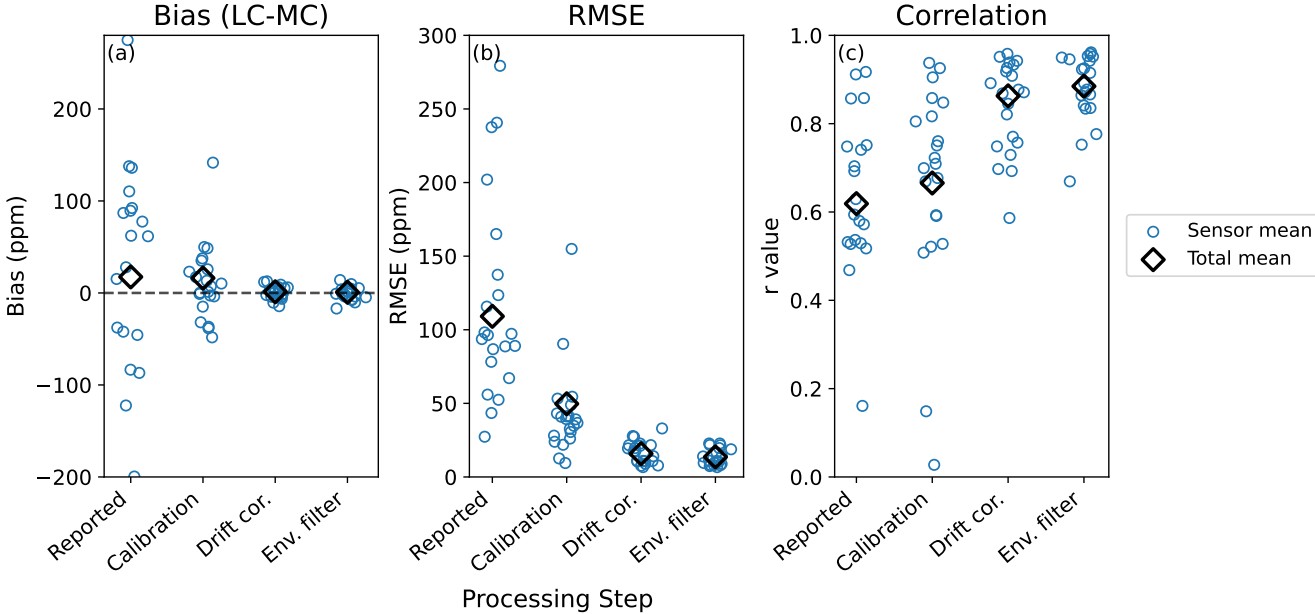

**Figure 6.** Error statistics of stepwise processing for low-cost $CO_2$ measurements when compared to parallel measurements of
reference mid-cost sensors at their field monitoring sites. Going from left to right in each subplot illustrates the improvement
after each processing step. Each point represents a single sensor, some of them collocated at the same site.

Finally, the data quality of the calibrated, drift-corrected and cleaned $CO_2$ concentration was assessed. The
performance was assessed at multiple levels: for the full network, per site, and per sensor. Compared to the mid-cost
network, the low-cost weighted mean network bias, RMSE, and $r$-value averaged over all observations at co-located
sites during the entire two-year study period are evaluated at 0.75±1.67 ppm, 13.61±1.38 ppm and 0.88 respectively,
with the uncertainty indicating the standard error. The means are weighted by the number of observations per
sensor. However, since the reference mid-cost sensors also have a known RMSE of 1 ppm (Grange et al., 2025), the
RMSE stated above is the RMSE of the low-cost and mid-cost sensors combined. After error variance subtraction of
the mid-cost RMSE for each low-cost sensor, we end up with a network mean RMSE of 13.57 ± 1.39 ppm. Averaged
per site, the values ranged between a bias of -3.81 and 9.42 ppm, RMSE of 8.3 and 18.5 ppm and $r$-value of 0.82 and
0.95, and for individual sensors between a bias of -16.81 and 14.12 ppm, RMSE of 6.6 and 22.7 ppm and $r$-value of
0.83 and 0.96, with one outlier of 0.67.





In addition, the final data quality was assessed per season (Fig. 7). Especially the bias shows a strong seasonal dependence, with the best performance in spring and fall. This is likely because the calibration models were trained in spring, resulting in best model performance under spring-time weather conditions. Some weather dependency of the drift correction methodology could also play a role. The largest bias was found during winter and summer, during which low-cost sensors respectively under- and overestimated the $CO_2$ concentration compared to the mid-cost sensors. On average, the yearly bias of all co-located sites was close to zero, indicating the network systematic error is small. However, for site averages, the bias still reached up to $\pm 9.4$ ppm, and for individual sensors up to $\pm 16.8$ ppm.

The RMSE showed slightly elevated values in summer and fall, when diurnal variations are generally larger, resulting in a larger data spread. The RMSE appears to be the largest when the fluxes are high (e.g. 'reck' in summer, and 'zsch' in winter).

The average yearly correlation ranging from 0.80 to 0.93 between sites (Fig. 7c) indicates that low-cost sensors generally follow the same patterns as the mid-cost sensors. For individual sensors, the spread is around the same range, with one outlier at 'zhbr'. Especially at the grassland site 'reck' with distinct $CO_2$ dynamics, the low-cost sensors are in agreement with the mid-cost sensors. However, at the forest site 'zhbr', there seem to be substantial differences in observed patterns, which is likely related to the sensor's placement near a building wall. The difference in correlation suggests that the placement of a sensor is crucial to its performance, and the sensor's are best placed in a ventilated location away from obstructions or other distinct sources.

At individual sensors, much stronger biases, larger RMSE's and lower $r$-values occur. When we would only look at the network uncertainty, these large differences per sensor would be neglected. By taking the average of the two paired sensors, some of the random error is ruled out and the data quality of the given site is improved. We will therefore use site averages for further analysis in Sect. 3.2 and 3.3.

The averaged time series of all co-located low-cost sensors shows that these sensors are in good agreement with the mid-cost sensors (Fig. 8). Again, the time series demonstrate an overestimation of the low-cost sensors in summer (by up to $+10$ ppm) and an underestimation in winter (by up to -15 ppm). However, since the timespan only entails two years, it is difficult to allocate this bias to a seasonal cycle. Individual sensor errors still fluctuate substantially between -50 and 50 ppm. Especially at the vegetated site 'reck' and the traffic site 'zsta', the low-cost sensors show large deviations from the mid-cost observations.

The measurement uncertainty of $13.6 \pm 1.4$ ppm implies that the ZiCOS-L network is suitable for detecting $CO_2$ fluctuations at scales larger than 13.6 ppm. This uncertainty of ZiCOS-L is slightly larger than the RMSE of 10 ppm found for the Carbosense network by Müller et al. (2020). A major difference with Müller et al. (2020) is that they evaluated the uncertainty with respect to Picarro reference analysers outside the city. In our study, the uncertainty is evaluated at six field sites in the city, which are directly exposed to local sources and sinks. Cai et al. (2025) found an RMSE of 6 ppm for their SenseAir K30 sensors, which are in a similar price class as our LP8 sensors. This substantially lower uncertainty was obtained in a setup at a single site where three sensors were measuring



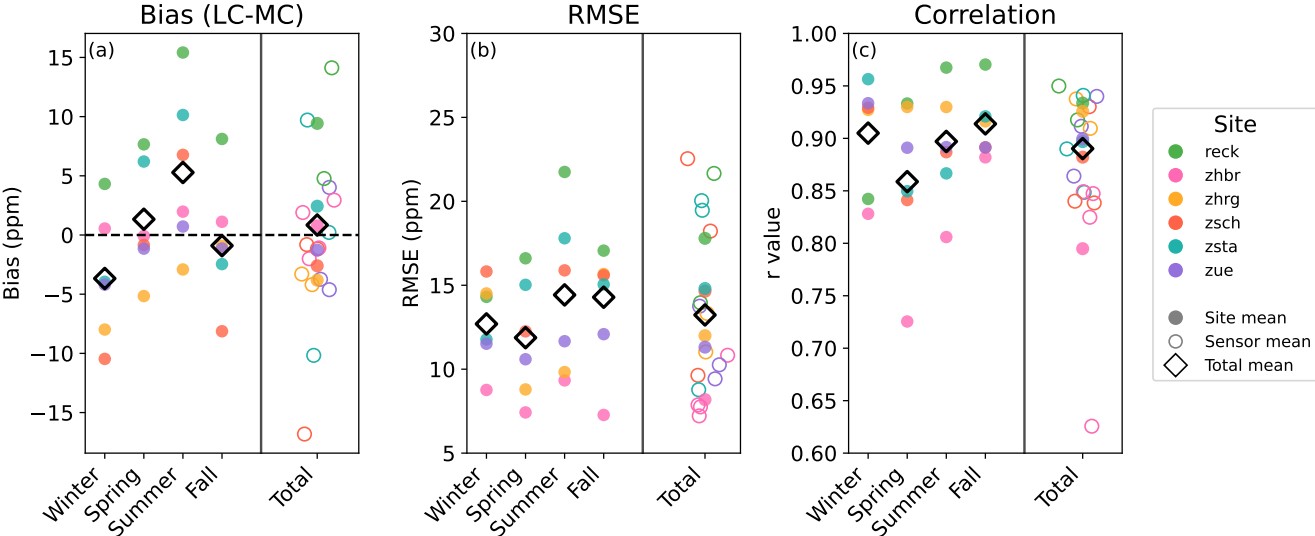

**Figure 7.** Error statistics for low-cost $CO_2$ sensors when compared to parallel measurements of reference mid-cost sensors at their field monitoring sites after the final processing step. The two years of observations are aggregated per season left of the vertical black line, and the total is shown right of this line. Filled dots show site means and empty circles show sensor means. Sample means of all sensors for each season or the total are presented by a black diamond. All averages are weighted averages, taking into account the number of observations (hourly) per sensor.

simultaneously inside a box in an indoor environment, but next to a window to simulate field conditions, which is thus not comparable with the true field evaluation in our study.

With an RMSE of $13.6 \pm 1.4$ ppm, ZiCOS-L clearly exceeds the extended compatibility goal of $0.2$ ppm for the interpretation of $CO_2$ data defined by the World Meteorological Organization (WMO) within their Global Atmo-
410  sphere Watch framework (World Meteorlogocial Organization, 2014). This compatibility goal is set for data usability for resolving regional gradients and for regional inversion modelling. However, the goal of ZiCOS-L is to provide insights into local and short-term $CO_2$ dynamics close to emission sources, which are often in the order of tens of ppm. Our results, thus, suggest that the network is suitable to detect $CO_2$ dynamics at high spatial and temporal resolution, despite the high measurement uncertainty.





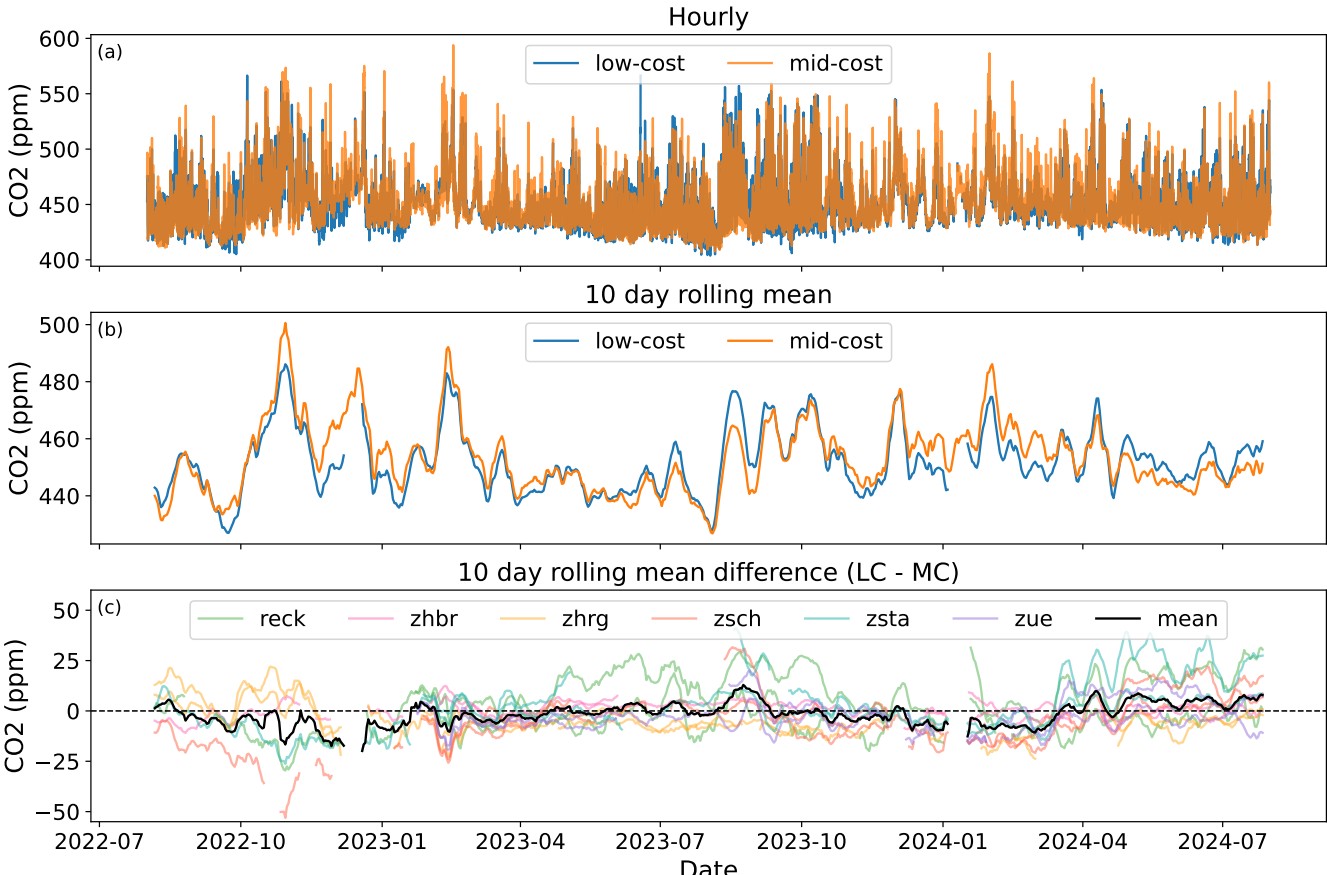

**Figure 8.** a) Hourly, and b) 10-day rolling mean time series averaged over all co-located sites comparing mid-cost (orange) and low-cost (blue) sensors. Panel c) shows the difference between each low-cost sensor and the mid-cost sensor at the same site, coloured by site, and in black the weighted mean of all co-located sensors.



## 3.2 CO₂ spatial gradients

The mean concentrations of the 56 sites in the ZiCOS-L network varied between 438 and 465 ppm (Fig. 9a, which is comparable to the values found for ZiCOS-M (Grange et al., 2025). Differences in mean concentrations between the sites are, to some extent, explained by environmental conditions of the sites. The lowest concentration was measured at the vegetated site 'zhbr', which is located at roughly 200 m above the city, adjacent to the Zürichberg forest (Fig. 1). Strikingly, some of the highest concentrations were also measured at vegetated sites ('reck', 'zbrc' and 'rctz'). Opposed to 'zhbr', these sites are in rather flat terrain and close to agricultural grassland where strong nighttime respiration causes high $CO_2$ accumulation. For better site comparisons, we take the mean afternoon concentrations (i.e. daily minima), when turbulent mixing is strongest and the sensors therefore are least influenced by local sources and sinks. The highest daily minima concentrations are measured close to the city centre (near Zürich Hauptbahnhof; main railway station) and at low altitude (Fig. 9b). There are some exceptions for traffic sites where high concentrations are measured outside the city centre or at higher altitude, and for some vegetated or unspecified sites that measure low concentrations close to the city centre.

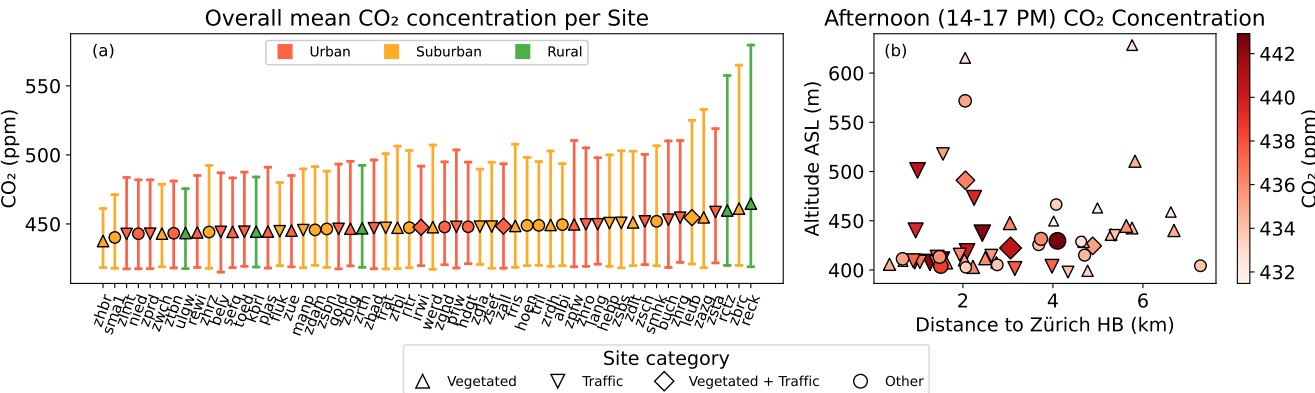

**Figure 9.** (a) Average $CO_2$ concentrations during the entire study period (August 2022 to July 2024) per monitoring site, with error bars indicating the 5th to 95th percentile. (b) Mean afternoon (14:00–17:00 local time) $CO_2$ concentrations at each site plotted against altitude above sea level (a.s.l.) and distance to Zurich Hauptbahnhof (HB). Colours and marker size indicate the afternoon $CO_2$ concentration. In both panels, marker shapes specify the site category.

Interpolated $CO_2$ concentration maps give insights into the spatial variability of $CO_2$ and allow for quick detection of sites with high concentrations (hotspots) in the city. For that, we used the mean afternoon (14:00–17:00 local time) concentrations at each site and applied an inverse distance weighting (IDW) interpolation to obtain concentration maps for each season (Fig. 10). Spatial gradients within the city were smallest during summer and spring (10 ppm) and largest in autumn and winter, when the differences in concentration were 16 and 18 ppm respectively (Fig. 10). Furthermore, the spatial plots display that some sites measure high concentrations year-round, while other sites only





show high concentrations during a specific season (Fig. 10). Especially during spring, some sites that show lower
concentrations in the other seasons show enhanced concentrations. Most interestingly, hotspots are predominantly
made up of traffic sites while vegetated sites typically act as a local sink.



**Figure 10.** Inverse Distance Weighting (IDW) interpolation of mean afternoon (14:00—17:00 local time) $CO_2$ concentration
per season.

### 3.3   $CO_2$ temporal dynamics

The previous section already showed that spatial patterns in $CO_2$ concentrations can, to some extent, be explained
by dominant sources or sinks in the surroundings such as vegetation or traffic. In this section, more detail is added
by analysing the different groups of vegetation type and density, traffic intensity and human activity based on the



classification framework described in Sect. 2.1. Diurnal cycles of the different classification groups are presented in Fig. 11.

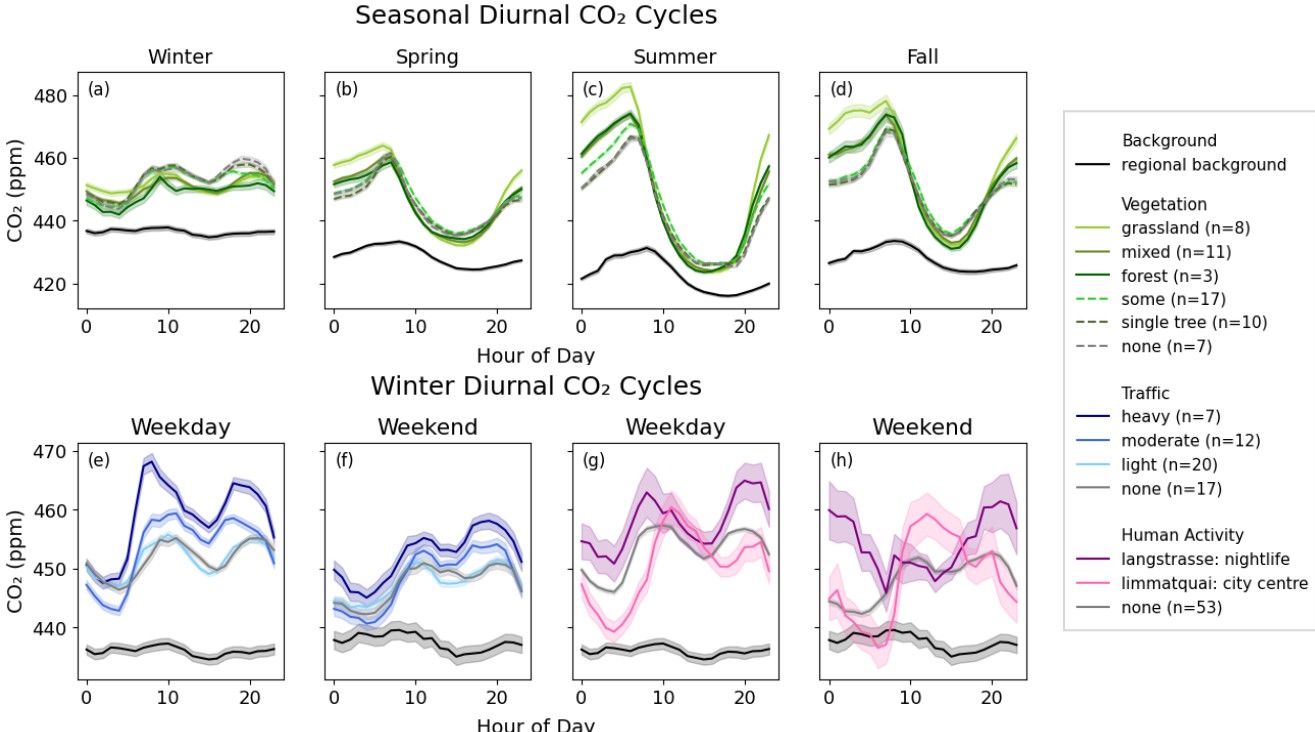

**Figure 11.** Diurnal $CO_2$ cycles grouped by: (a–d) vegetation, (e, f) traffic intensity, and (g, h) human activity. The top row shows vegetation classes split by season; solid lines represent densely vegetated sites (grassland, mixed, or forest), dashed lines indicate sparsely vegetated sites (some, single tree, or none). The bottom row shows traffic sites and sites influenced by human activity split by weekdays and weekend days, using winter data only. Shading indicates variability across sites (5th – 95th percentile). For human activity, two individual sites are selected and shading indicates variability at the site.

General seasonal and diurnal patterns in $CO_2$ concentration can be deduced from the regional background measured at selected sites around the city (Appendix B). The background signal in Fig. 11a–d shows that the diurnal range is smallest in winter and largest in summer. During the growing season, and especially in summer, the background concentration peaks in the morning due to biogenic respiration combined with a confined boundary layer, and is reduced to the daily minimum in the afternoon (14:00–15:00) by increased atmospheric dispersion and biogenic uptake (Crawford et al., 2016). In winter, when the vegetation is less active, the concentration is high due to the combination of higher emissions, a confined boundary layer and little uptake by vegetation. Winter diurnal $CO_2$ cycles in the city are generally dominated by anthropogenic emissions with peak concentrations during morn-





ing and afternoon rush hour (Ward et al., 2015; Velasco and Roth, 2010). In the background concentration, these anthropogenic effects are diluted.

Vegetation dynamics locally amplify general seasonal patterns. The densely vegetated sites (Fig. 11a–d, solid lines) experience strongly elevated morning peaks and nighttime concentrations during the growing season, whereas for
sparsely vegetated sites (Fig. 11a–d, dashed lines) this enhancement is much less pronounced. This is known as the rectifier effect, which is caused by covariance in boundary layer dynamics and biogenic $CO_2$ fluxes. Nighttime respiration in a stable atmosphere accumulates at the surface, while the effect of daytime $CO_2$ absorption is diluted due to strong vertical mixing (Denning et al., 1999). This results in an apparent $CO_2$ source in the morning, while over the entire day, the site is a net sink. This effect was already observed in the ZiCOS-M network for a single
vegetated site (Grange et al., 2025), and is now confirmed for various vegetation types and densities thanks to the large number of sites in the ZiCOS-L network.

The rectifier effect is largest in summer when the morning respiration peak is around 10 ppm higher than at non-vegetated sites (Fig. 11c). The effect is even stronger for grassland (around 20 ppm), which is consistent with the higher respiration rates over lawns compared to tree sites in the city of Zurich reported by Stagakis et al. (2025).
The sparsely vegetated sites are in agreement with each other, but in summer, the sites with 'some' vegetation experience a slightly higher ($< 5$ ppm) morning peak and nighttime concentration than the sites without vegetation. Another interesting observation is that sites with a single tree in proximity to the sensor closely follow the pattern of sites without vegetation, suggesting that the biogenic effects of a single tree on the surrounding concentration are negligible.

To explore effects of anthropogenic activities, the data is split by weekday (Monday to Friday) and weekend (Saturday and Sunday), as this strongly affects behavioural patterns of citizens (Fig. 11e and f). Because the vegetation effect is dominating the concentration levels during the rest of the year, the effects of anthropogenic activities were analysed only during winter months when the vegetation is less active. Figure 11e shows that during winter weekdays, sites with heavy and moderate traffic intensity show distinct $CO_2$ peaks during the morning (7:00–
9:00 local time) and evening (16:00–19:00 local time) rush hours, corresponding to typical commuting times. These peaks are amplified due to boundary layer confinement before sunrise and after sunset. The largest concentration increase due to traffic was found at heavy traffic sites during the morning rush hour peak, resulting in a concentration enhancement of 15 ppm compared to sites with light or no traffic. At sites with moderate traffic intensity, the traffic caused $CO_2$ concentrations of around 5 ppm higher throughout the day. Such a response to rush hour traffic was also
found by Shusterman et al. (2018), who showed a strong correlation between local $CO_2$ concentration and traffic flow during the morning rush hour in the BEACO$_2$N network in San Francisco. During weekends, leisure travel caused the morning rush hour peak to spread over a broader time window, while the evening peak remained distinct, despite being less intense than during the week.

The busy nightlife street 'Langstrasse' shows elevated concentrations compared to the average of the other sites
throughout the monitoring period. Figure 11h shows the largely elevated concentration at nighttime during the





weekend. The results suggest that nightlife at this site causes a local $CO_2$ enhancement of around 15 ppm due to human activity (including traffic). During both weekdays and weekends, we see evening (18:00-22:00) concentrations elevated by around 10 ppm. It should be noted that Friday evening is counted as 'weekday', even though it is expected to reflect weekend behaviour, whereas for Sunday evening it is the other way around.

Interestingly, concentrations at 'Limmatquai' show a different diurnal pattern than all other sites, with $CO_2$ concentration peaking around noon. This can be linked to the large number of daytime visitors in the city centre of Zurich and is caused by both traffic and human respiration. Additionally, Fig. 11g shows that this effect is similar throughout the week and thus not enhanced during the weekend. Overall, Fig. 11 demonstrates that low-cost sensors effectively capture temporal variations that are characteristic of certain types of emissions, at a profoundly detailed

and local scale.

### 3.4    Case study: Public events

In addition to the generalised spatial patterns, hotspots and temporal dynamics due to different sources, the low-cost sensors are also able to detect certain public events. In this section, we analyse two events: Züri Fäscht and Zurich Street Parade.

Züri Fäscht is a two-and-a-half-day summer festival occurring every three years in the city centre and around Lake Zurich, with the last edition held on Friday, July 7 until Sunday, July 9 2023. In 2023, the festival attracted an estimated two million visitors and included music stages, fairground activities and fireworks. Due to environmental reasons, the traditional air squadrons show has been suspended since 2019, and some of the fireworks were replaced by drone shows in 2023. During the festival, there is a 72-hour car curfew in the city centre, and most of the city

centre turns into a pedestrian zone (Tages-Anzeiger, 2023a).

     Zurich Street Parade is an annual techno music parade around Zurich's lake basin, drawing nearly one million visitors to the city. The route starts on the east side of the lake and goes along the lake basin, stopping by several music stages and finishes on the west side of the lake (Tages-Anzeiger, 2023b).

     Both the Streetparade and Züri fäscht draw large crowds, which goes hand in hand with elevated human respiration

emissions. Figure 12 shows the spatial distribution and temporal dynamics of the $CO_2$ concentration during these events. The maps (Fig. 12a and d) show the $CO_2$ anomalies compared to the mean summer $CO_2$ concentrations during the same time of day. For both events, the event location is consistent with dark red areas in the concentration maps (Fig. 12a and d), indicating $CO_2$ concentrations up to 45 ppm higher than mean summer concentration. What specifically stands out during Züri Fäscht, is the strong local decrease in concentration (Fig. 12a, highlighted in blue)

at a site just outside the event area. This is the traffic site 'zsta' where $CO_2$ concentration (Fig. 9) is usually high due to traffic emissions. During the car curfew implemented during Züri fäscht, the site experienced a concentration of around 32 ppm lower than during other summer evenings. A large part of this signal is likely caused by the traffic reduction, but since meteorological conditions are not investigated, they might also play a role. The car curfew in





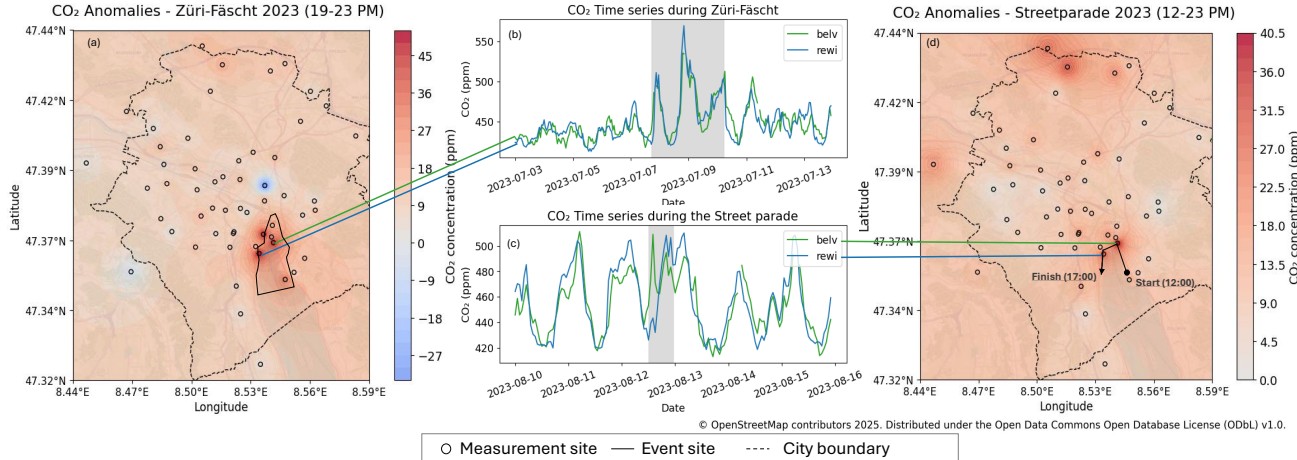

**Figure 12.** $CO_2$ anomalies during two public events in Zurich: Züri Fäscht (panel a, b) and Street Parade (panel c, d). Maps in panel (a) and (d) show spatially interpolated mean $CO_2$ concentrations during the events normalized against the mean summer concentration during the same event hours. Panel (b) and (c) show time series at two sites in the event areas, with shaded areas marking the event duration.

the city centre also suggests that almost all of the $CO_2$ enhancement can be attributed to human respiration, aside

from some other sources like the fair attractions or fireworks.

In addition to the concentration maps, the $CO_2$ time series measured at two sites at the event locations also show elevated concentrations during the time of the event (Fig. 12b and c). During Züri fäscht, the concentration peaks on Saturday evening, which is the busiest night of the event with two firework shows. A closer look at the $CO_2$ time series during the Street Parade, in combination with the parade route, reveals extremely local and short-term

fluctuations corresponding to the passage of the parade. Firstly, a distinct peak can be detected at 'belv', near the start of the route, followed by a peak at 'rewi', which is nearing the end of the route. These two case studies suggest that the ZiCOS-L network is well suited to detect local and short-term events.

## 4    Conclusions

In this study, we have described the design and deployment of the ZiCOS-L network and outlined the data processing

steps required to obtain the final $CO_2$ data. The sensor performance has been evaluated and insights into the spatial variability and temporal dynamics of the network have been presented.

The measurement performance of the ZiCOS-L network was assessed by comparing it to the reference ZiCOS-M network at six co-located sites. After several processing steps including calibration, drift correction and filtering, the network bias was assessed at $0.75\pm1.67$ ppm, and the RMSE at $13.6\pm1.4$ ppm at the mid-cost sensor sites. For

individual sensors, the bias ranged between -16.8 and 14.1 ppm, and the RMSE between 6.6 and 22.7 ppm. Averaged



per site, some of the random error is removed and the values ranged between a bias of -3.8 and 9.4 ppm and an RMSE of 8.3 and 18.5 ppm. For further analyses, we decided to use the site averages. An alternative to taking site means would be to select the best performing sensor based on a performance evaluation prior to deployment. Additionally, we would recommend future networks to deploy low-cost sensors in triplets instead of pairs, to facilitate the detection

of malfunctioning sensors.

The mean $CO_2$ concentration during the study period (August 2022 to July 2024) varied from 438 to 465 ppm between sites and differences can largely be explained by the sites' surroundings. Higher altitude and larger distance to the city centre correlated with lower concentrations, whereas vegetation, traffic and human activity largely influenced the $CO_2$ level at some sites, independent of altitude and distance to the city centre. Spatial interpolation between

the sites allowed for the detection of $CO_2$ hotspots. During the afternoon, when the atmosphere is well mixed, all local hotspots were found at traffic sites, while most vegetated sites acted as a sink.

A detailed classification of vegetation type and density, traffic intensity, and human activity demonstrated the network's ability to effectively capture local processes. Densely vegetated sites had an amplified rectifier effect of around 10 ppm in summer, with an additional 10 ppm for grassland sites where respiration is even stronger. The

effect of a single tree on the local $CO_2$ concentration was shown to be negligible. On the other hand, heavy traffic sites clearly measured a peak in concentration during morning rush hours of up to 15 ppm higher than at sites with light or no traffic. Lastly, at sites with a distinct pattern of human visitors, the $CO_2$ concentration closely followed the behavioural pattern of the people, resulting in high concentrations during peak visitor hours.

Besides the spatial variability and the temporal dynamics due to local processes, the network was also able to

detect specific events. Two major public events in the city of Zurich (Züri fäscht and the Street Parade) were shown to locally enhance $CO_2$ concentrations at the time of the event. During Züri fäscht, when a car curfew was implemented, a strong local $CO_2$ decrease due to reduced traffic was also observed. Localized $CO_2$ dynamics could also be detected during the Street Parade, causing short-term concentration peaks as the parade passed by two of the measurement sites.

Most interestingly, the detailed insights into local $CO_2$ dynamics detected by the low-cost network are much smaller than 13.6 ppm. This suggest that averaging over time or sites brings down the uncertainty considerably. As a result, useful information can be extracted from the low-cost network at a scale well below the determined network uncertainty of 13.6 ppm.

Our study showed that low-cost networks like ZiCOS-L can be a cost-effective alternative to high-cost networks

to obtain information about $CO_2$ dynamics in a city. However, our results also suggest that useful observations can only be obtained after careful data processing, which requires the use of higher-cost measurement equipment. The combination of these different tiers of monitoring systems brings up the total cost. However, extending existing networks with low-cost sensors can be an affordable way to add insights into local dynamics.

While the ZiCOS-L network does not achieve the precision required to meet the extended compatibility goal as

defined by the WMO (World Meteorlogocial Organization, 2014), it provides valuable insights into the spatial and





temporal variability of $CO_2$ in the city of Zurich. The data analyses provided information about distinct emission patterns associated with various surrounding such as vegetation or traffic, which is crucial to better understand $CO_2$ sources and sinks within the city. Moreover, these findings are also relevant for city stakeholders because they offer valuable information on $CO_2$ hotspots or the impact of urban vegetation. Such information is essential to support

cities in reducing GHG emissions and implementing their climate action plans. The usefulness of the ZiCOS-L observations in estimating city emissions within an atmospheric inversion framework will be investigated in a future study.





## Appendices

### Appendix A: Site names

Table A1: Overview of all low-cost sensor site, stating their site abbreviation, sitename, site category, site area type, spatial location and elevation (a.s.l.)

| site | site full name | site category | site area | latitude | longitude | elevation |
|------|----------------|---------------|-----------|----------|-----------|-----------|
| albi | Albisriederdörfli | Traffic | suburban | 47.375445 | 8.485062 | 429.6 |
| belv | Bellevueplatz | Traffic | urban | 47.36733 | 8.54427 | 407.7 |
| buch | Bucheggplatz | Traffic | urban | 47.398067 | 8.533654 | 473.2 |
| fluk | Kirche Fluntern | Other | suburban | 47.376665 | 8.559698 | 517.8 |
| frat | Frankental Höngg | Traffic | suburban | 47.406174 | 8.481087 | 435.2 |
| fris | Friedhof Sihlfeld | Vegetated | suburban | 47.376353 | 8.506247 | 411.9 |
| gold | Goldbrunnenplatz Wiedikon | Traffic | suburban | 47.370362 | 8.514015 | 418.9 |
| hdgt | Hardgutstrasse | Other | urban | 47.38529 | 8.50414 | 405.2 |
| hebp | Hegibachplatz | Traffic | suburban | 47.361984 | 8.560806 | 437.4 |
| hoen | Meierhofplatz Höngg | Other | suburban | 47.402851 | 8.499296 | 466.5 |
| irwi | Irchelpark | Both | urban | 47.396261 | 8.545072 | 491.1 |
| kbrl | Breitloostrasse Kilchberg | Vegetated | suburban | 47.325834 | 8.537545 | 510.5 |
| lang | Langstrasse | Traffic | urban | 47.378964 | 8.526948 | 408.6 |
| leub | Leutschenbach | Vegetated | suburban | 47.418815 | 8.563832 | 424.5 |
| manp | Manesseplatz Wiedikon | Traffic | suburban | 47.365711 | 8.521442 | 415.5 |
| nied | Niederdorf Hirschenplatz | Other | urban | 47.37317 | 8.54383 | 411.3 |
| pfiw | Pfingstweidstrasse Hardturm | Traffic | urban | 47.392454 | 8.503088 | 401.5 |
| plas | Platzspitz | Vegetated | urban | 47.381507 | 8.539678 | 405.8 |
| rctz | Katzensee | Vegetated | rural | 47.434222 | 8.506875 | 440.1 |
| reck | Reckenholz | Vegetated | rural | 47.427839 | 8.517464 | 442.6 |
| rewi | Rentenwiese | Vegetated | urban | 47.363611 | 8.5368 | 407.4 |
| ritr | Oerlikon Riedgrabenweg | Other | suburban | 47.408618 | 8.558584 | 425.7 |
| sefq | Seefeldquai Chinagarten | Vegetated | urban | 47.354748 | 8.550541 | 407.8 |
| sma1 | Fluntern Krähbühlstrasse | Other | suburban | 47.378333 | 8.566353 | 571.8 |
| smhk | Schlieren Mühleackerstrasse | Other | suburban | 47.394408 | 8.445823 | 404.3 |
| toed | Tödistrasse Enge | Traffic | urban | 47.365992 | 8.534961 | 408.4 |
| tril | In der Ey Triemli | Other | suburban | 47.371064 | 8.490861 | 431.6 |
| ulgw | Ringlikon Langwiesstrasse | Vegetated | rural | 47.357384 | 8.469371 | 628.4 |
| werd | Blaue Werdinselbrücke | Vegetated | suburban | 47.399959 | 8.484743 | 399.2 |
| zall | Allmendstrasse Brunau | Traffic | urban | 47.352494 | 8.524758 | 422.3 |





Table A1: Overview of all low-cost sensor site, stating their site abbreviation, sitename, site category, site area type, spatial location and elevation (a.s.l.)

| site | site full name | site category | site area | latitude | longitude | elevation |
|------|----------------|---------------|-----------|----------|-----------|-----------|
| zazg | Schule Auzelg | Vegetated | suburban | 47.413856 | 8.57215 | 428.6 |
| zbad | Badenerstrasse Altstetten | Traffic | urban | 47.387394 | 8.488266 | 403.8 |
| zblg | Bullingerhof | Vegetated | urban | 47.379055 | 8.51228 | 410.4 |
| zbrc | Birchstrasse Seebach | Vegetated | suburban | 47.425738 | 8.542765 | 436.0 |
| zdam | Dammweg Sihlquai | Other | suburban | 47.388802 | 8.526714 | 404.0 |
| zdlt | Döltschiweg Wiedikon | Vegetated | suburban | 47.365775 | 8.503296 | 447.6 |
| zfbl | Eichrain Seebach | Vegetated | suburban | 47.42826 | 8.55042 | 444.7 |
| zghd | Gerhardstrasse Wiedikon | Other | urban | 47.370449 | 8.52326 | 413.3 |
| zgla | Gladbachstrasse Oberstrass | Traffic | suburban | 47.383309 | 8.549973 | 501.1 |
| zhbr | Heubeeribüel | Vegetated | suburban | 47.381469 | 8.565916 | 615.4 |
| zhrgc | Rosengartenstrasse (container) | Traffic | urban | 47.39517 | 8.5261 | 432.7 |
| zhro | Herostrasse Altstetten | Other | urban | 47.393928 | 8.48667 | 397.9 |
| zhrz | Schule Hirzenbach Schwammendingen | Other | suburban | 47.403619 | 8.587727 | 428.8 |
| zlmt | Limmatquai Helmhaus | Other | urban | 47.369227 | 8.543297 | 408.3 |
| zpfw | Pfingstweid | Vegetated | urban | 47.388008 | 8.513537 | 403.1 |
| zprd | Paradeplatz | Other | urban | 47.370135 | 8.539081 | 409.4 |
| zrdh | Schule Riedenhalden | Vegetated | suburban | 47.418834 | 8.511141 | 463.0 |
| zrth | Rütihofstrasse Oberengstringen | Vegetated | suburban | 47.411963 | 8.466966 | 458.9 |
| zsbn | Stampfenbrunnerstrasse Altstetten | Other | suburban | 47.385854 | 8.477936 | 415.1 |
| zsbs | Seebahnstrasse Aussersihl | Traffic | suburban | 47.378328 | 8.519075 | 409.9 |
| zsch | Schimmelstrasse | Traffic | urban | 47.37098 | 8.523579 | 413.0 |
| zsef | Schule Seefeld | Traffic | suburban | 47.357166 | 8.555189 | 414.5 |
| zsta | Stampfenbachstrasse | Traffic | urban | 47.386738 | 8.539802 | 440.2 |
| ztbn | Turbinenplatz | Other | urban | 47.389714 | 8.51776 | 402.7 |
| zue | Zürich Kaserne | Other | urban | 47.377576 | 8.530397 | 408.6 |
| zwch | Wachtelstrasse Wollishofen | Vegetated | suburban | 47.343035 | 8.527064 | 449.7 |

**Appendix B: Background sites**

The names and locations of the reference analysers outside the Zurich city boundaries are shown in Fig. B1. The background concentration depicted in Fig. 11 is composed of the concentration of either 'Lägern Hochwacht', 'Birchwil Turm' or 'Beromünster', depending on the wind direction.



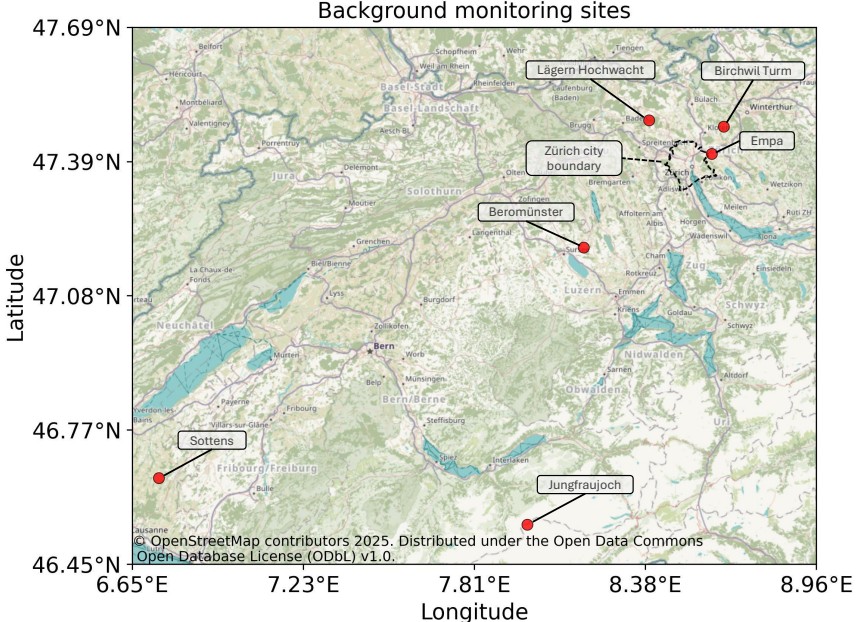

**Figure B1.** Zoomed-out map showing the reference sites part of the network outside the Zurich city area.

## Appendix C: Climate chamber testing

Extensive climate chamber experiments have been conducted to determine environmental factors that influence the final measurement, response times and offset concentration values of the sensors. The goal was to characterize the individual sensors before deployment in the field. The sensors were put in a climate chamber (Feutron KPK800) equipped with an additional custom valve controller able to control $T$ (0–60 °C $T$), $RH$ (0–100 %) and $CO_2$ (range between 400 ppm and 4000 ppm).

A calibrated, high precision reference instrument (Picarro G2401) was used to control the $CO_2$ in the climate chamber and also to compare the signals against. The climate chamber was equipped with a ventilator that ensured a well-mixed atmosphere within.

A measurement cycle (between each change in parameter) was always separated into the two phases "startup" and "measurement". Normally, the "startup" phase lasted approx. 3 h if $T$ and $RH$ were changed (and 6 h if $CO_2$ concentration was

changed) and the "measurement" phase normally lasted 3 h. Only one parameter was changed in between measurements, and at least 4 different set points (in a range deduced by normal to extreme real-world environmental occurrences) per parameter were measured. Additionally, to measure the effects of different absolute humidity values, experiments were conducted where $T$ and $RH$ set points were changed at the same time to explore the humidity effects with various different absolute humidities also experienced during field deployment.

$T$ was varied in a range between 2 °C to 32 °C in intervals of 10 °C. Absolute humidity $RH$ was changed in a range of 5 g $H_2O$/kg$_{\text{dry air}}$ to 14 g $H_2O$/kg$_{\text{dry air}}$ in intervals of 3 g $H_2O$/kg$_{\text{dry air}}$. $RH$ was changed in a range between 35 % to 93 %





in intervals of 19 %. $CO_2$ was changed in a range of 400 to 700 ppm in intervals of 75 ppm $CO_2$. Further measurements were conducted with dynamic changes of temperature, humidity and $CO_2$ concentrations to emulate different daily cycles at different seasonalities. $T$ and $RH$ were cycled in a sine profile while the $CO_2$ was cycled in a cosine profile. This pattern was

reflecting the fact that under environmental conditions, $CO_2$ is normally low (afternoon) when $T$ and $RH$ is high. The cycle was run with a period of 12 h, halved after two complete cycles, down to a period of 3 h. The speed-up was carried out to determine the sensors' step responses. Zero air experiments to determine the sensors' response to the absence of $CO_2$ were tried but not conducted successfully, as the climate chamber could not be evacuated and flushed properly to achieve near-zero $CO_2$ concentration values. As near-zero $CO_2$ concentrations do not occur in environmental conditions, this experiment was

not pursued further.

During evaluation of the climate chamber dataset it was found that the real environmental data (during co-location of the sensors with a reference instrument) proved to be more reliable for initial baseline calibration of the sensors, as described in the sections above. Therefore, the climate chamber data was not used any further for sensor calibration. Nevertheless, the climate chamber experiments allowed a profound analysis and exploration of the sensors' behaviour to changes of different

environmental parameters if changed isolated but also combined.

## Appendix D: Drift correction

Figure D1 presents a schematic of the drift correction described in Sect. 2.4.4.

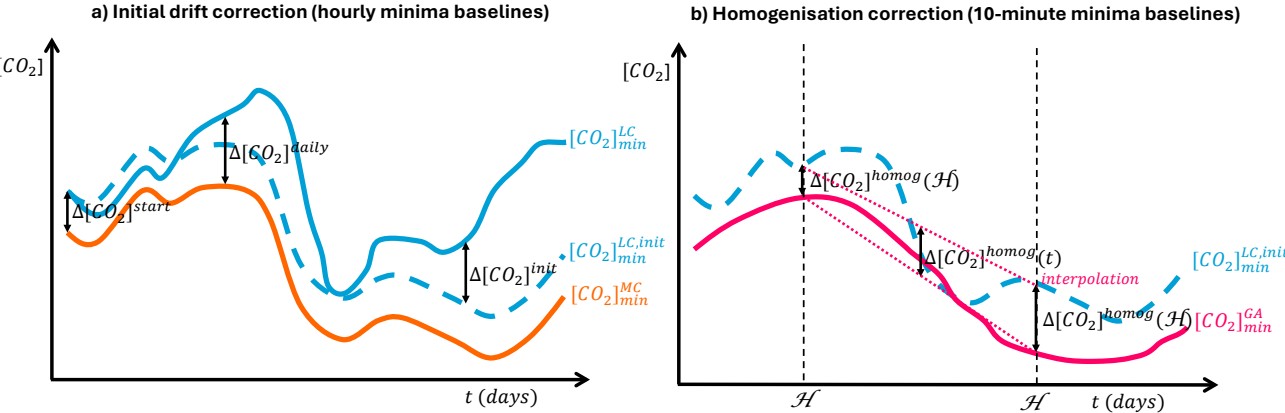

**Figure D1.** A schematic of how a typical low-cost sensor (in blue) suffers from drift over time and how it is corrected. The solid lines indicate the baseline (daily minima) of the low-cost observations (blue), extracted mid-cost prediction (orange) and reference gas analyser (pink). The multiple components of the offset coefficient (Equation 6) are indicated with black arrows. Panel (a) shows the initial correction towards the mid-cost network baseline, in which the baselines follow the minimum hourly concentration. Panel (b) shows how the initial correction is updated during homogenous days, in which the baselines follow the minimum 10-minute concentration.



*Data availability.* The mid-cost data and the calibrated and drift corrected low-cost data are available via the ICOS Cities data portal (https://citydata.icos-cp.eu/portal). Low-cost data for which the additional step of environmental filtering was applied are available at C (Creman, 2025).

*Author contributions.* LC performed the data analysis and prepared the manuscript, figures and tables. SKG designed the methodology, conducted the data processing and prepared a first draft of the methodology part of the manuscript including some figures. LB and LC were responsible for the planning and execution of the data analysis and the manuscript preparation. LB, DB, CH and LE planned and designed the manuscript. SKG, PR and LE installed and maintained the measurement network. All authors contributed to the interpretation of the results and the manuscript revision.

*Competing interests.* At least one of the (co-)authors is a member of the editorial board of Atmospheric Measurement Techniques.

*Acknowledgements.* We would like to thank the environmental office of the City of Zurich (UGZ) and Swisscom for their support with sensor deployment, and Decentlab GmbH for the sensor's hardware and data infrastructure. We thank Simone Baffelli, Lucas Fernandez Vilanova and Yuri Brugnara for their efforts in maintaining and improving software systems. $CO_2$ data from three additional NABEL and ICOS Switzerland (ICOS-CH) sites were kindly provided by Beat Schwarzenbach and Stephan Henne. The continued operation of the sensor network would not have been possible without the dedicated assistance of civil service volunteers and we acknowledge the contributions of Wisnu Lang, Simon Rohrbach, Davide Bernasconi, Hannes Wäckerlig, Michael Kovac, Josua Stoffel, Urban Brunner, Ulysse Schaller, Leonardo Beltrami (a visiting student), Stefan Lampart, Yann von der Weid, Jan Krummenacher, Quirin Beck, Max Geniets, Joshua Bourquin and Gianluca Poretti. Finally, we would like to thank the ICOS Cities sensor network teams in Munich and Paris for the productive collaboration. We also acknowledge the use of artificial intelligence tools to assist with coding for data analysis and to compute figures. The authors retain full responsibility for all results and their interpretation.

*Financial support.* This work is part of ICOS Cities, also know as PAUL, Pilot Applications in Urban Landscapes - Towards integrated city observatories for greenhouse gases, which has received funding from the European Union's Horizon 2020 Research and Innovation Programme under grant agreement no. 101037319. ICOS Switzerland (ICOS-CH) monitoring activites are funded by the Swiss National Science Foundation (grant no. 20F120_198227).



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
