# Peer review of "The Zurich Low-cost CO2 sensor network (ZiCOS-L): data processing, performance assessment and analysis of spatial and temporal CO2 dynamics"

_EGUsphere, 2025_

## Referee Comment (RC1)

General comments:

This paper reported a low-cost sensor $CO_2$ network consisting of 56 sites of Zurich. The sensors require in-field training for model calibration and further post-processing steps to calibrate drifts and exclude outliers. Validated against parallel reference measurements from the mid-cost sensor network, the hourly mean root mean squared error was 13.6±1.4 ppm and the mean bias 0.75±1.67 ppm, which is a reasonably good performance. I think this study falls in the scope of AMT, but the MS needs further substantial revisions for potential publication in this journal.

Major comments:

1. The environmental corrections are linear, while for factors of temperature and pressure, the responses can be non-linear (in Section 2.4), and air pressure seems not included in the processing, more discussions and tests are needed.

2. NDIR sensors are sensitive to vibrations during potential rough handling (e.g. transport, and setup), this may be the reason why climate chamber models not suitable for filed calibration;

3. The LP8 sensors may have a SenseAir automatic baseline correction (ABC) algorithm for the manufacture product raw

accuracy, need to close it before the authors' model calibrations. Potential jumps may be seen in Fig.8b and 8c, peaks and valleys (e.g. around 2022-10; 2023-01; 2023-09; 2024-02). Please introduce this in Section 2.4.2;

4. In line 188, do the 3-12 weeks long enough to cover all typical T, RH, P variability through the year? Especially for the relative short 3 weeks;

5. In line 316, the authors excluded temperatures below 0 ℃, please provide the fractions deleted by this factor, this may not be a major problem for this study, but for many places in winter, this problem largely reduced the sensor use areas and needs further development in algorithm or adding heater;

6. In lines 374-376, need to point out and discuss the shortcomings of larger summer and winter biases in further atmospheric inversions;

7. In lines 403-406, the published paper of Cai et al., (2025) reported their final corrected RMSE of 1.6 ppm (daily) to 2.4 ppm (30-month) in the abstract, need to update the reference (https://amt.copernicus.org/articles/18/4871/2025/) and parameters.

8. In lines 431-432, for some sites and some seasons, the RMSE/Bias can be larger than the spatial gradients signal, and it needs to be

cautious when explain certain signals and further use in inversions;

9. Consider add some analyses or discussions on local or public inventories (ODIAC), e.g. in section 3.2, in Fig. 10 (correlations of high concentrations and emissions?);

10. The writing and expression need further improvements in the fluency and grammar errors.

Minor comments:

Keep consistent in figure title capitalization.

Line 126, may be not suitable to say commercially available to avoid potential advertisement?

Line 144-145, can be deleted.

Section 2.3 and 2.4 both have data processing?

Line 147, can briefly introduce tests in the climate chamber.

Line 152, update Cai et al., (2025) for published version.

Line 185-186, if climate chamber tests are not useful, may be not to include them in future works?

Line 203, be replaced by median value or set null, which is more reasonable?

Line 206, "aggregation" is better replaced with "averaged".

Line 207-208, why not used in this paper?

Line 225, introduce the reference gas (e.g. accuracy).

Line 237, show/describe the typical differences of second and lowest?

Line 242, show same cases of the almost identical ones in supporting information?

Line 285, also stronger sink by vegetation uptake in summer.

Line 302, **linearly-**interpolated?

Fig.6c, why correlation become worse after calibration for outlier?

Line 379-380, seems there are temperature impacts.

Line 425-427, add some explanations.

Line 430, IDW first appeared in Line 234-235;

Line 434-435, add typical sites.

Fig.10 caption, add time periods.

Fig.11 Add shading areas as in Fig.12 to show rush hours.

Fig.11 change "fall" to "autumn" to keep consistency with previous figures.

Line 449 higher emissions from heating?

Line 483 week to weekdays?

Line 546, no industrial or other emissions? Explore inventories if possible.

Line 550, negligible at certain distance?

---

## Author Comment (AC1)

Reply to reviewer #1

We would like to thank the reviewer for the critical examination of the manuscript and the valuable suggestions. The points will be addressed one by one, with the reviewer comments depicted in black, the author reply in blue and the alterations to the manuscript in green.

General comments:

This paper reported a low-cost sensor $CO_2$ network consisting of 56 sites of Zurich. The sensors require in-field training for model calibration and further post-processing steps to calibrate drifts and exclude outliers. Validated against parallel reference measurements from the mid-cost sensor network, the hourly mean root mean squared error was 13.6±1.4 ppm and the mean bias 0.75±1.67ppm, which is a reasonably good performance. I think this study falls in the scope of AMT, but the MS needs further substantial revisions for potential publication in this journal.

Major comments:

1. The environmental corrections are linear, while for factors of temperature and pressure, the responses can be non-linear (in Section 2.4), and air pressure seems not included in the processing, more discussions and tests are needed.

    We thank the reviewer for raising the awareness that responses can indeed be non-linear. However, please note that multiple models were tested, including both (multi-)linear and non-linear models (line 172). Air pressure and specific humidity were also included in these tests, but after testing we concluded that the linear model including only T and RH performed best. Note that these parameters are directly measured by the sensor units. A key reason for the linear model is its robustness outside the conditions encountered during the calibration period. Although a longer calibration period would make it possible to cover a wider range of conditions, this approach has clear limitations, and measurements outside the calibration range cannot be avoided. In fact, we deliberately refrained from searching for highly elaborate models that might yield slightly better results during the calibration phase and at the calibration site because such models risk overstating their performance under real-world conditions, that is, at other locations, under different environmental conditions, and potentially outside the calibration range.

2. NDIR sensors are sensitive to vibrations during potential rough handling (e.g. transport, and setup), this may be the reason why climate chamber models not suitable for filed calibration

We thank the reviewer for this critical comment. This could indeed explain part of the deviation. However, we are confident there must be several other factors at play. Multiple works have shown similar results indicating that climate chamber models are not suitable for low-cost sensors, as mentioned in line 186.

3. The LP8 sensors may have a SenseAir automatic baseline correction (ABC) algorithm for the manufacture product raw accuracy, need to close it before the authors' model calibrations. Potential jumps may be seen in Fig.8b and 8c, peaks and valleys(e.g. around 2022-10; 2023-01; 2023-09; 2024-02). Please introduce this in Section 2.4.2

   The algorithm was turned off prior to any data processing. Therefore, we think it is unnecessary to address. The potential jumps are likely caused by local changes in environmental conditions.

4. In line 188, do the 3-12 weeks long enough to cover all typical T, RH, P variability through the year? Especially for the relative short 3 weeks

   We agree that the 3-12 weeks calibration time is relatively short, and therefore seasonal variability is not fully captured. However, this is a general shortcoming for many LC sensor networks. Our choice for a linear model does help reduce the error when extrapolating outside the calibration period. In previous work (Müller et al., 2020, AMT) we combined climate chamber calibration (covering a very wide range of conditions) and field tests in the calibration model. During the long and nationwide field measurements, this did not significantly improve performance. Nevertheless, we agree that the short calibration time still is a shortcoming that needs be addressed. Therefore we added the following to the manuscript in line 192-194:

   "We acknowledge that the calibration period is too short to capture local and seasonal variability. However, the use of a linear model reduces the sensitivity to short-term fluctuations and facilitates a more robust extrapolation outside the calibration period"

5. In line 316, the authors excluded temperatures below 0 ℃, please provide the fractions deleted by this factor, this may not be a major problem for this study, but for many places in winter, this problem largely reduced the sensor use areas and needs further development in algorithm or adding heater

   We agree that this is incomplete. We added the rejection rate of each filtering step separately, and the combined rejection rate. The point about the constraint for other regions is also included:

"For our data, this resulted in a rejection rate of 4.8%. However, in colder climatic regions, a similar threshold would limit the amount of observations significantly." in line 321

"…resulting in good data quality with a rejection rate of 13%." in line 339

"The total rejection rate after both filtering steps is 17%." in line 341

6. In lines 374-376, need to point out and discuss the short comings of larger summer and winter biases in further atmospheric inversions

We must note that further atmospheric inversions are outside the scope of this paper, which focusses solely on low-cost measurements. However, to put the seasonality of the biases in perspective, we added line 541-543 to the conclusion:

"Biases also showed a seasonal dependence. Inter-sensor and inter-seasonal variability in bias should be carefully considered when interpreting the results and for further use of the data."

7. In lines 403-406, the published paper of Cai et al., (2025) reported their final corrected RMSE of 1.6 ppm(daily) to 2.4ppm(30-month) in the abstract, need to update the reference(https://amt.copernicus.org/articles/18/4871/2025/) and parameters.

We thank the reviewer for pointing this out, the reference and number have been updated.

8. In lines 431-432, for some sites and some seasons, the RMSE/Bias can be larger than the spatial gradients signal, and it needs to be cautious when explain certain signals and further use in inversions

Indeed, the large RMSE/Bias should be taken into account when interpreting results, as mentioned in line 399-400. However, we strongly believe that the strength of this network is the possibility to aggregate measurements of multiple sensors in both space and time. Our results confirm this, because intuitive patterns can be detected at scales smaller than the measurement error. This indicates that a significant part of the error is random and averaging can even out the error to some extend, as explained in line 560-563. We agree that this should be mentioned earlier in the manuscript. Therefore we added:

"Additionally, averaging measurements can even out part of the random error." In line 418

9. Consider add some analyses or discussions on local or public inventories (ODIAC), e.g. in section 3.2, in Fig. 10 (correlations of high concentrations and emissions?)

ODIAC would not be the right resolution. Brunner et al. (2025, https://doi.org/10.5194/acp-25-14387-2025) used MapLuft to show that most hotspots are caused by traffic emissions. They showed that heating emissions are the largest source but very homogenously spread, and emissions by industry are small within the city. The low-cost sensors are shown to detect emissions at very small street-level scale (like traffic) and can therefore not be related to more regional emissions from heating or industries. Therefore, only traffic is included by use of site categories. For completeness, we added:

"In line with findings of Brunner et al. (2025), local effects of industries and heating were found to be negligible." In line 442-443

10. The writing and expression need further improvements in the fluency and grammar errors.

The paper has been checked again in fluency, and grammar errors have been corrected.

Minor comments:

Keep consistent in figure title capitalization. Line 126, may be not suitable to say commercially available toavoidpotential advertisement?

➔ Deleted

Line 144-145, can be deleted.

➔ Since the Zicos-L and Zicos-M paper are largely linked, we think it is important to mention the similarity.

Section 2.3 and 2.4 both have data processing?

➔ The title of section 2.3 is changed to "Data transmission and storage", because the processing is described in section 2.4.

Line 147, can briefly introduce tests in the climate chamber.

➔ Since climate chamber tests are not used, the further explanation can be found in the appendix.

Line 152, update Cai et al., (2025) for published version.

➔ Updated

Line 185-186, if climate chamber tests are not useful, may be not to include them in future works?

➔ We agree that future field studies with LC sensors can omit climate chamber testing, as this paper adds to the list of work which show that it is not representative for field conditions. However, climate chambers are very useful during the process of developing LC sensors to quickly identify and quantify their response to a multitude of conditions, including cross-sensitivities.

Line 203, be replaced by median value or set null, which is more reasonable?

➔ We thank the reviewer for pointing out this mistake. The value was set to null, the text has been altered.

Line 206, "aggregation" is better replaced with "averaged".

➔ "aggregation" has been changed to "averaging"

Line 207-208, why not used in this paper?

➔ For Zicos-M, the Hampel identifier was used for a different purpose. This has been accentuated in the text: "The ZiCOS-M network also utilized a more strictly tuned Hampel identifier, but for the different purpose of identifying possible local contamination events instead of removing outliers."

Line 225, introduce the reference gas (e.g. accuracy).

➔ This is done in the following 2 sentences. Accuracy of the MC network is included.

Line 237, show/describe the typical differences of second and lowest?

➔ A one year timeseries of daily minima from the lowest site, second-lowest site and median of the remaining sites are depicted in the following figure. The next figure describes typical difference between lowest and second-lowest site daily minima.

[Figure]

[Figure]

Line 242, show same cases of the almost identical ones in supporting information?

➔ Showing and comparing these three dimensional surfaces is very difficult. However, there was visually no difference between the surfaces, and the two methodologies also resulted in virtually the same performance statistics. Besides, this is a minor step necesary to sepperate the correction from the validation, and including 3D figures would be too much information for such a minor step.

Line 285, also stronger sink by vegetation uptake in summer.

➔ This does not necessarily cause larger gradients in winter/autumn. In the afternoon, antropogenic emissions are stronger than biospheric fluxes as shown in Stavros et al. (2025, https://doi.org/10.5194/bg-22-2133-2025) and Brunner et al. (2025, https://doi.org/10.5194/acp-25-14387-2025).

Line 302, linearly-interpolated?

➔ Yes it is linearly interpolated, see Appendix D. "linear" has been added to the sentence.

Fig.6c, why correlation become worse after calibration for outlier?

➔ The axes limit hides one of the outliers.

Line 379-380, seems there are temperature impacts.

→ Boundary layer dynamics, refer to midcost paper for these dynamics (fig 8&9). Larger range/variability which causes larger RMSE. No changes to text

Line 425-427, add some explanations.

→ Text has been altered: "The highest daily minima concentrations are generally measured close to the city centre (near Zürich Hauptbahnhof; main railway station) and at low altitude, where there are more pollution sources and the background concentration is higher (Fig 9b). However, daily minima are most dominantly affected by site category."

Line 430, IDW first appeared in Line 234-235;

→ Change made

Line 434-435, add typical sites.

→ Sites gold, zsch and buch are added to the text.

Fig.10 caption, add time periods.

→ Time period added

Fig.11 Add shading areas as in Fig.12 to show rush hours.

→ Shading is added for weekdays rush hours

Fig.11 change "fall" to "autumn" to keep consistency with previous figures.

→ Fall is changed to autumn

Line 449 higher emissions from heating?

→ Anthropogenic emissions are dominant over biogenic uptake/emissions. Heating falls under anthropogenic emissions.

Line 483 week to weekdays?

→ Change made

Line 546, no industrial or other emissions? Explore inventories if possible.

→ The magnitude of industrial emission sources is known to be relatively small in the city of Zurich (Brunner et al., 2025). Additionally, heating sources mainly affect the background concentration and thus do not cause hotspots.

Line 550, negligible at certain distance?

→ The category "single tree" indicates a tree in the vicinity of the sensor. This has been added to the text.

---

## Author Comment (AC2)

Reply to reviewer #2

In line with the reply to reviewer #1, the reviewer comments will be depicted in black and the author reply in blue

This paper presents a comprehensive discussion of the deployment, calibration and performance of a dense array of low-cost sensors. The paper is interesting, shows that the sensors have potential for scientific application and is clearly written. The initial analyses explaining differences in local concentration patterns was very insightful and interesting. I enjoyed reading this paper. I recommend publication after clarification of some minor issues.

We would like to thank the reviewer for this positive feedback, and would like to address the minor issues in the next section.

I found the discussion of humidity slightly confusing. Humidity has at least two effects. First water contributes to the total pressure requiring a correction if one is expressing a dry air mixing ratio. Second water affects the spectroscopy of CO2 in the NDIR sensor both through pressure broadening and through spectral interference. The authors begin by saying they make a correction to dry air. I was confused about when in the sequence of their analysis they make the correction to dry air and whether it makes any difference if they do that at the beginning or end of their analysis. I'm not challenging what the authors are doing, just hoping for some clearer procedural description.

The conversion to dry air is done as a very first step, as described in section 2.4.1. We used the term "water dilution correction" for this, in line with the ZiCOS-M paper. However, this might be misleading, so we changed the title of the section to "conversion to dry air mole fraction". The spectroscopic effect of humidity is tackled in section 2.4.2, by fitting the multi-linear model. The concentration first needs to be converted to dry air, before testing the sensitivity to RH.

Second, is more subtle. A comparison of calibrated data to a reference should be flat if plotted against RH, absolute humidity or temperature. It's unclear to me if the authors checked their calibration in this way. A brief comment on the residuals of the low cost sensors vs. the reference along these coordinates would be helpful in evaluation of the data set.

We thank the reviewer for this sharp comment. The residual plots before and after calibration are added to this document. We see that after Calibration, the residual plots of most sensors are flat and close to zero. However, Random errors of the low-cost sensors still cause a large spread. Additionally, the hockey curve behaviour adressed in section 2.4.5 can be spotted for a few sensors (e.g. 1251, 1311, 1144) for RH, that indicates the large error of LC sensors for high RH. For T, some of the sensors still show some non-linear T dependency after calibration, which could not be captured by our linear model. Additionally, it should be noted that LC and MC do not measure at the

exact same location. MC sensors are mounted much higher to rooftops while LC sensors measure at street level. Differences in T and RH between these levels can also explain part of the residuals.

[Figure]

**CO₂ residuals (LC − MC) vs T**
**Dry conversion vs Calibration**

[Figure]

Temperature (°C)

**CO₂ residuals (LC − MC) vs q**
**Dry conversion vs Calibration**

[Figure]

Specific humidity (kg/kg)